# Nanoparticles and Antiviral Vaccines

**DOI:** 10.3390/vaccines12010030

**Published:** 2023-12-27

**Authors:** Sen Liu, Meilin Hu, Xiaoqing Liu, Xingyu Liu, Tao Chen, Yiqiang Zhu, Taizhen Liang, Shiqi Xiao, Peiwen Li, Xiancai Ma

**Affiliations:** 1Guangzhou National Laboratory, Guangzhou International Bio-Island, Guangzhou 510005, China; liu_sen@gzlab.ac.cn (S.L.); hu_meilin@gzlab.ac.cn (M.H.); liu_xiaoqing@gzlab.ac.cn (X.L.); xingyu.liu693@gmail.com (X.L.); chen_tao01@gzlab.ac.cn (T.C.); zhu_yiqiang@gzlab.ac.cn (Y.Z.); liang_taizhen@gzlab.ac.cn (T.L.); xiao_shiqi@gzlab.ac.cn (S.X.); li_peiwen@gzlab.ac.cn (P.L.); 2School of Biology and Biological Engineering, South China University of Technology, Guangzhou 510006, China; 3State Key Laboratory of Respiratory Disease, Guangzhou Medical University, Guangzhou 511400, China; 4Zhongshan School of Medicine, Sun Yat-Sen University, Guangzhou 510080, China

**Keywords:** nanoparticle, vaccine, adjuvant, viral infection, innate immunity, adaptive immunity

## Abstract

Viruses have threatened human lives for decades, causing both chronic and acute infections accompanied by mild to severe symptoms. During the long journey of confrontation, humans have developed intricate immune systems to combat viral infections. In parallel, vaccines are invented and administrated to induce strong protective immunity while generating few adverse effects. With advancements in biochemistry and biophysics, different kinds of vaccines in versatile forms have been utilized to prevent virus infections, although the safety and effectiveness of these vaccines are diverse from each other. In this review, we first listed and described major pathogenic viruses and their pandemics that emerged in the past two centuries. Furthermore, we summarized the distinctive characteristics of different antiviral vaccines and adjuvants. Subsequently, in the main body, we reviewed recent advances of nanoparticles in the development of next-generation vaccines against influenza viruses, coronaviruses, HIV, hepatitis viruses, and many others. Specifically, we described applications of self-assembling protein polymers, virus-like particles, nano-carriers, and nano-adjuvants in antiviral vaccines. We also discussed the therapeutic potential of nanoparticles in developing safe and effective mucosal vaccines. Nanoparticle techniques could be promising platforms for developing broad-spectrum, preventive, or therapeutic antiviral vaccines.

## 1. Introduction

Pathogenic viruses have long been great threats to public health. Historically, numerous pandemics have been caused by viruses, such as the Spanish flu caused by H1N1 in 1918, Ebola hemorrhagic fever caused by the Ebola virus in 1976, acquired immunodeficiency syndrome (AIDS) caused by the human immunodeficiency virus (HIV) in 1981, avian influenza caused by H5N1 in 1996, severe acute respiratory syndrome (SARS) caused by SARS-CoV in 2002, Middle East respiratory syndrome (MERS) caused by MERS-CoV in 2012, and coronavirus disease 2019 (COVID-19) caused by SARS-CoV-2 in 2019 [1]. Each emerging infectious disease causes tens of thousands of deaths and exerts a significant impact on global health. More zoonotic viruses gradually cross-transmit to humans and spread within human society, which brings great threats to public health and may potentially trigger the next wave of global pandemic. Therefore, searching for effective drugs and vaccines against these pathogenic viruses is still urgently needed.

### 1.1. Major Pathogenic Viruses and Their Pandemics

The Spanish flu, which occurred in 1918, was caused by the H1N1 influenza virus and belongs to the category of A-type influenza viruses (Figure 1). It resulted in approximately 50 million deaths between 1918 and 1919 [2,3]. The influenza virus belongs to the *Orthomyxoviridae* family of RNA viruses, which is categorized into types A, B, C, and D [4]. The A-type influenza virus is prone to antigenic mutations and is the most common type of influenza virus during the flu season. The efficacy of influenza vaccines is typically around 40–60% and varies each year depending on the matching degree between the antigens covered by the vaccine and the prevalent influenza virus strain [5,6]. Although the protection provided by influenza vaccines is limited, they are still regarded as important preventive measures against influenza infection.

SARS, MERS, and COVID-19 are caused by SARS-CoV, MERS-CoV, and SARS-CoV-2, respectively, which are highly pathogenic coronaviruses of the *Orthocoronavirinae* subfamily of the *Coronaviridae* family (Figure 1) [7]. Compared with SARS-CoV-2, SARS-CoV is less contagious but more lethal [8,9]. According to the World Health Organization (WHO), a total of 8096 cases of SARS infection had been reported globally in 2002–2003, resulting in 774 deaths. MERS-CoV has limited infectiousness and is spread mainly through close contact with infected persons. According to the WHO, over 2500 MERS cases and around 860 deaths have been reported globally since 2012. COVID-19 has caused a worldwide pandemic with profound impact and loss on global health and socioeconomic development. As of November 2023, over 771 million people have been infected by SARS-CoV-2, resulting in 6.97 million deaths. Apart from the three highly pathogenic coronaviruses, there are four human coronaviruses with lower pathogenicity, which are HCoV-229E, HCoV-OC43, HCoV-NL63, and HCoV-HKU1 [10]. Infection with these four coronaviruses only causes mild symptoms and does not cause large-scale transmission but still exhibits strong pathogenicity for infants and elderly individuals.

Arboviruses, including Dengue virus (DENV), Zika virus (ZIKV), yellow fever virus (YFV), West Nile virus (WNV), and Rift Valley fever virus, are predominantly transmitted to humans via mosquitoes, while RVFV is mainly transmitted by fleas (RVFV) [11]. DENV is one of the most notorious arboviruses and was listed as one of the top ten threats to global health in 2019. DENV-mediated Dengue fever is mainly prevalent in tropical areas [12]. The prevalence of Dengue fever has constantly increased over the past few decades, causing 50–100 million infections and 20,000 deaths each year, imposing a huge socioeconomic burden on global public health [13]. Zika fever is another viral infectious disease caused by ZIKV [14]. In 2015–2016, an outbreak of Zika fever occurred in South America, causing millions of infections. During the ZIKV epidemic in Brazil, ZIKV was found to be related to severe complications of the nervous system, such as Guillain–Barré syndrome and neonatal microcephaly [14]. Ebola hemorrhagic fever, primarily occurring in West Africa and equatorial regions, is a severe infectious disease caused by the Ebola virus of the *Filoviridae* family [15]. According to information released by the WHO regarding the Ebola virus disease, the infection has affected over 15,000 individuals from 1976 to 2020 and has a high fatality rate of 75% due to a lack of effective treatments, including antiviral drugs and vaccines.

In contrast to typical acute viral infections, chronic viral infections are characterized by their long-term infection, which is hard to eliminate. Human immunodeficiency virus (HIV), hepatitis B virus (HBV), and hepatitis C virus (HCV) all can cause chronic infection. HIV, a member of the *Retroviridae* family, is categorized into two major subtypes, including HIV-1 and HIV-2, based on their genetic characteristics [16]. HIV-1 is the most prevalent subtype globally and the primary cause of HIV infection. Although 40 years have passed since the first case of AIDS, HIV/AIDS remains a significant global public health issue, having claimed the lives of millions of people to date [17]. Unfortunately, no complete cure or effective vaccine is available for HIV infection. HBV is a DNA virus of the *Hepadnaviridae* family [18]. According to WHO data, around 200 million people worldwide live with chronic HBV infection, with around 90% of patients being infected at birth or during infancy. Asia, Africa, and Pacific Island countries are areas with high HBV prevalence. With the introduction and promotion of HBV vaccines, many Asian countries have successfully lowered the infection rate of HBV [19]. Distinct from HBV, HCV belongs to the *Hepacivirus* genus of the *Flaviviridae* family [20]. The global number of people infected with HCV is approximately 58 million. Fortunately, significant progress has been made in treating HCV through next-generation direct-acting antivirals (DAAs); however, an effective anti-HCV vaccine is still unavailable [21].

### 1.2. Antiviral Vaccines and Adjuvants

Antiviral drugs and vaccines are the most effective strategies to prevent and cure viral infections. However, most drugs fail to eradicate viruses or provide long-term protection, while drugs against HCV have been able to allow the organism to get rid of the virus. Vaccination remains the most effective and sustainable method to prevent infectious diseases [22]. Currently, preventive and therapeutic vaccines mainly include inactivated vaccines, nucleic acid vaccines, viral vector vaccines, and recombinant protein vaccines [23,24,25,26,27,28]. Different types of vaccines possess distinct advantages, the effectiveness of which can be further enhanced upon co-administration with appropriate adjuvants.

Inactivated vaccines are in vitro-cultured pathogens that have been treated utilizing physical or chemical methods, such as formalin or β-Propiolactone, to eliminate their biological activity while preserving their antigenicity (Figure 2) [29,30]. Inactivated vaccines have a well-established history of application with mature and stable manufacturing processes. Their research and development are less time consuming and have been applied to prevent multiple diseases [31]. Although inactivated vaccines are potentially safe to use, the manufacturing processes may compromise antigenic components and decrease the preventive effectiveness of these vaccines. As a result of their low immunogenicity, inactivated vaccines can only maintain protective immunity for a short duration and need a high dose of inoculation with multiple immunizations.

In 1990, scientists discovered that when DNA or mRNA with genetic information was injected directly into mice, the expression of the corresponding proteins could be detected in cells, proving the feasibility of nucleic acid vaccines [32]. Over the past few decades, a surge of developing nucleic acid vaccines has been sparked in the field of antiviral research. Nucleic acid vaccines include DNA vaccines and mRNA vaccines (Figure 2). DNA vaccines function by delivering expression vectors containing the DNA of antigen proteins into the human body, synthesizing the antigens in somatic cells, and thereby triggering immune responses to produce neutralizing antibodies (nAbs), as well as cell-mediated immune responses [33,34]. Without the need to synthesize antigens in vitro, DNA vaccines are easy to manufacture and safe to use, and the vaccines themselves can also possess inherent adjuvant properties, reducing the cost and making them suitable for mass production. However, a potential drawback is the risk of randomly integrating into the host genome. Foreign DNAs are also susceptible to degradation within the human body and need to pass through several barriers entering the cell nucleus, resulting in a reduced immunogenicity of DNA vaccines [35].

mRNA vaccines share a similar mechanism as DNA vaccines, as both function by synthesizing antigens within the human body. However, unlike DNA vaccines, mRNA vaccines do not need to enter the cell nucleus for expression and exhibit superior delivery efficiency, leading to higher immunogenicity [36]. Moreover, mRNA vaccines are also potentially safe since they do not contain viral protein components and have a shorter development cycle because synthesis and purification of antigen proteins are not required. Consequently, mRNA vaccines against viral mutants can be quickly developed to prevent infection. Because exposed mRNAs are not stable within the human body and are susceptible to degradation by RNase, carriers, such as liposomes, are commonly used to enhance delivery efficiency [37,38]. Since the outbreak of SARS-CoV-2 in 2019, mRNA vaccines have evolved rapidly and played a pivotal role in preventing SARS-CoV-2 infection. Currently, The Food and Drug Administration (FDA) has approved mRNA vaccines from Pfizer, BioNTech, and Moderna for market release [39]. Nonetheless, mRNA vaccines also have certain drawbacks, such as stringent transport and storage conditions, reported side effects in clinical applications, and partial muscle necrosis induced by vaccine inoculation [40,41].

Viral vector-based vaccines elicit immune responses utilizing genetically modified vector viruses, which are constructed by inserting the DNA sequence of antigens from pathogenic viruses into the genome of the harmless vector virus (Figure 2) [27]. The antigens will be expressed by human cells infected with vector viruses, and an immune response will be triggered. The major advantages of viral vector vaccines are minimal side effects and high gene delivery efficiency. Commonly used vector viruses include vaccinia virus, adenovirus, vesicular stomatitis virus, and influenza virus [42,43,44,45]. However, one challenge in developing viral vector vaccines is the pre-existing immunity. Antibodies targeting the vector virus may also be induced after vaccination, thereby weakening the effectiveness of the vaccine [46,47].

Recombinant protein vaccines involve the in vitro expression and purification of antigen proteins, the genes of which are cloned into expression vectors and expressed using *E. coli.*, yeast, or mammalian cells [28,48]. These vaccines encompass monomer subunit vaccines, dimer vaccines, trimer vaccines, and polymeric vaccines. The technology for developing recombinant protein vaccines is well-established and stable, ensuring a high level of safety. Moreover, these vaccines are readily recognized by immune cells and effectively activate host immune responses, demonstrating the virus prevention effect [49,50]. However, the expression yield of these recombinant protein vaccines is not as high as nucleic acid vaccines or viral vector-based vaccines. Additionally, the residue host-derived components, including nucleic acids, proteins, or endotoxins, may result in potential side effects, which need to be strictly monitored during large-scale production.

The aforementioned vaccines have played a pivotal role in preventing various viral infections; however, they still have certain limitations that compromise their efficacy in clinical applications [51,52]. Nanoparticle vaccines represent next-generation vaccine technologies, the nano-carriers of which include liposomes, polymers, inorganic nanoparticles, virus-like particles, and self-assembling protein nanoparticles. These ingenious-sized nanoparticle vaccines are more likely to be enriched in immune organs, such as spleens and lymph nodes, facilitating their recognition and processing by immune cells to elicit augmented innate and adaptive immune responses. Moreover, these vaccines employ nanoparticles as carriers of antigens, effectively safeguarding the antigens from degradation and increasing their stability in vivo [53,54]. The early protein nanoparticle vaccines utilized as display platforms were virus-like particles (VLPs), which are highly structured protein particles formed by the self-assembly of multiple structural proteins of viruses and are highly consistent with the morphology of natural viruses [55]. Due to their small size, VLPs can be effectively recognized by antigen-presenting cells, thereby activating the immune system and acting as a self-adjuvant [56]. The protein self-assembling nanoparticle vaccine has good biocompatibility and high homogeneity and can simultaneously display multiple antigens of different viruses or viral mutants. Therefore, it has great potential in designing broad-spectrum vaccines [57]. However, similar to the traditional recombinant protein vaccines, the expression yield of self-assembling proteins is relatively low, and there may be contaminants from host cells, which still present numerous challenges in the development process. Therefore, they have promoted the production of self-assembling protein nanoparticle vaccines from non-viral sources. It has been reported that gold nanoparticles (AuNPs), carbon nanotubes, silica particles, polymers, and liposome nanoparticles can elicit cytokine and antibody responses [58,59]. The nanoparticles generated by these substances can be served as carriers to protect and deliver antigens or act as adjuvants to enhance the immune efficacy of vaccines [60,61]. Nevertheless, the clinical application of nanoparticle vaccines is still in its nascent stage. The antigen selection is the expression system, and the nanoparticle size has the potential to impact the stability and immune responses elicited by these nanoparticle vaccines. Consequently, the optimal forms of vaccine delivery including nanoparticle-based approaches still need further exploration.

Adjuvants play crucial roles in antiviral vaccines. They are non-specific immune enhancers that can augment the magnitude and duration of the body’s immune response [62,63]. Additionally, adjuvants enhance the stability and immunogenicity of antigens and protect the antigens from hydrolysis. The incorporation of adjuvants can make the antigens more efficiently recognized by the immune system, thereby strengthening the innate and adaptive immune responses against viruses. Furthermore, their inclusion extends the effective duration and potency of the vaccine [64,65]. Adjuvants can also reduce the cost of vaccines by minimizing the usage amount of antigens and the number of vaccinations. The aluminum adjuvant is the first FDA-approved human vaccine adjuvant and the most widely employed immune adjuvant in clinical applications. The aluminum adjuvant enables a controlled and sustained release of antigens, thereby increasing the potency of vaccines by inducing robust humoral immune responses. Apart from the aluminum adjuvant, only six other adjuvants have been approved by the FDA for use in vaccines due to considerations such as safety, stability, effectiveness, and feasibility for large-scale production [66,67]. With the advancements in vaccines, more research on promising adjuvants is indispensable for future endeavors.

Above all, many different kinds of vaccines and adjuvants have been extensively developed and successfully implemented in clinical applications. Notably, among all of these vaccine platforms, nanoparticle-based next-generation techniques exhibit immense potential to confront pathogenic viruses, although their clinically technical routes are still under investigation. In the subsequent sections of this review, we will comprehensively summarize recent advancements in nanoparticle-based vaccines, particularly focusing on their applications in both acute and chronic viruses-targeted immunizations.

## 2. The Applications of Nanoparticles in Influenza Vaccines

Influenza is an acute respiratory infection caused by the influenza virus, which often occurs in the form of seasonal epidemics. It is a highly infectious and rapidly spreading disease that seriously threatens human life and health [68,69]. Influenza viruses are spherical, elliptical, or filamentous in shape, with diameters ranging from approximately 80 nm to 120 nm. Influenza viruses belong to the *Orthomyxoviridae* family, which encompass pathogens affecting both animals and humans. These viruses contain single-stranded, negative-chain, and segmented RNAs as their genomes. According to types of viral nucleoprotein (NP) and matrix protein (MP), influenza viruses are classified into types A, B, C, and D [70]. The predominant viruses that currently infect humans are influenza A and B viruses [71,72,73]. Based on combinations of viral surface proteins hemagglutinin (HA) and neuraminidase (NA), influenza A viruses contain H1N1, H3N2, H5N1, H7N9, and many other subtypes. Influenza A viruses have led to waves of influenza pandemics, such as the 1918–1920 Spanish flu (H1N1), which claimed tens of millions of people worldwide, the 1957–1958 Asian flu (H2N2), which caused 2 million deaths, and the 1968–1970 Hong Kong flu (H3N2), which resulted in an estimated one to four million human lives [74,75,76]. In 2009, a novel influenza A (H1N1) virus caused a pandemic, which was the first major influenza outbreak in the 21st century. Although exhibiting less severe symptoms compared to previous pandemics, it still caused tens of thousands of deaths [77]. Influenza B viruses do not exhibit distinct subtypes, and currently, there are two circulating influenza B virus lineages named Yamagata and Victoria [78]. Compared with influenza A virus, influenza B virus is typically less pathogenic and commonly does not cause a pandemic. The transmission route of the influenza virus occurs through coughing, sneezing, and the exhalation of droplets and small particles containing the virus, entering the body via the respiratory tract, or indirect contact with the source of infection [79]. The incubation period of the influenza virus is very short, lasting only 1–2 days. Moreover, it is highly contagious and can quickly spread within the population. Influenza viruses predominantly infect respiratory mucosal epithelial cells, leading to host cell degeneration, necrosis, and subsequent shedding of host cells. This process causes mucosal congestion, edema, and heightened secretion levels, resulting in nasal congestion, a runny nose, sore throat, dry cough, and other symptoms of upper respiratory tract infection [80]. In individuals with compromised immunity or underlying diseases, severe symptoms and even a risk of death may occur after contracting influenza viruses [81].

Vaccination represents the most efficacious approach for preventing influenza virus infection and significantly mitigating the risk of influenza-related serious complications. Moreover, nanomaterials provide more options for vaccine delivery and display systems [82]. Both organic and inorganic nanoparticles have been employed in the development of influenza vaccines, as well as adjuvants (Figure 3, Table 1). Organic nanoparticles include polymer nanoparticles, protein nanoparticles, and virus-like nanoparticles (VLPs) [83]. To date, a variety of polymer nanoparticles have been extensively used in drug delivery research. Polymer nanoparticles are highly desirable materials for the nano-delivery of drugs due to their convenient synthesis, diverse structures, excellent biocompatibility, and superior biodegradability. However, due to the insufficient levels of mucosal IgA and cellular immune responses in the respiratory tract induced by the existing inactivated swine influenza A virus (SwIAV) vaccine, current intramuscular-injected inactivated SwIAV vaccines failed to provide protection against heterologous viral mutants. Chitosan is a kind of natural cationic polymer which can easily bind to mucosal surfaces. Thus, a chitosan-containing delivery system is an ideal technique for delivering drugs or vaccines to induce protective mucosal immunity [84]. Santosh Dhakal et al. employed chitosan polymer-based nanoparticles (CNPs) as a vaccine delivery platform to encapsulate killed SwIAV H1N2 antigen (KAg) [85]. After intranasal immunization with CNPs-encapsulated KAg (CNPs-KAg), vaccinated pigs exhibited elevated levels of antigen-specific IgG antibodies in sera and mucosal IgA antibodies in nasal swabs, which significantly reduced nasal viral shedding and lung viral titers upon heterologous influenza virus challenge. Similarly, another FDA-approved biodegradable polymer named polylactate-glycolic acid (PLGA) also showed significant enhancement of vaccine-induced protective immunity. The intranasal vaccination of PLGA-encapsulated inactivated SwIV H1N2 antigens (PLGA-KAg) in pigs increased the proliferation of antigen-specific lymphocytes and enhanced the proportions of functional T-helper, memory T, and cytotoxic T cells [86]. Additionally, PLGA-KAg provided potent protection against both homologous H1N2 and heterologous H1N1 SwIV challenges. These results indicate that vaccines co-delivered with CNPs or PLGA-NPs can elicit cross-protective immune responses against influenza viruses. Hydrogels are polymer materials with three-dimensional network structures, which have attracted much attention in the fields of biomedical engineering, biotechnology, and many others. Gillie A. Roth et al. constructed a polymer–nanoparticle (PNP) hydrogel vaccine delivery platform using dynamic multivalent non-covalent interactions between polymers and nanoparticles (NPs) [87]. The PNP hydrogels are formed by mixing TLR7/8 agonist-conjugated poly(ethylene glycol)-b-poly(lactic acid) (PEG-PLA) NPs with dodecyl-modified hydroxypropylmethylcellulose (HPMC-C_12_) polymers. PNP hydrogels enabled efficient and sustained co-delivery of TLR7/8 agonist adjuvants and influenza A H1N1 HA antigens to improve the efficiency of lymph node (LN) targeting, thereby reducing the risk of systemic exposure to adjuvants and associated toxicity. Most importantly, PNP hydrogels enhanced the durability and breadth of influenza subunit vaccines, which have the potential to confront future influenza strains. Taken together, polymer nanoparticles are promising influenza vaccine delivery systems for enhancing the efficacy of both vaccines and adjuvants.

Protein nanoparticles are newly emerged antigen delivery and display platforms, and ferritin is the most widely used self-assembling protein in both basic research and clinical trials (Figure 3, Table 1). The non-haem ferritin nanoparticle derived from *Helicobacter pylori* (*H. pylori*) can self-assemble into 24 stable polymers and has been successfully applied in influenza nanoparticle vaccines. Masaru Kanekiyo et al. genetically inserted the extracellular domain of influenza virus A/New Caledonia/20/1999 (1999NC) HA into the *H. pylori* ferritin sequence to express HA-ferritin protein nanoparticles in mammalian cells [88]. Compared to licensed inactivated trivalent influenza vaccines, HA-ferritin nanoparticles induced a stronger immune response in vivo, thereby enhancing the potency and breadth of specific antibody responses. Recently, two ferritin nanoparticle vaccines based on the influenza HA domain (H2 subtype) and the H1 stabilized stem region have completed Phase I clinical trials (NCT03186781 and NCT03814720). The trial results demonstrated that H2HA-ferritin nanoparticles were potentially safe and characterized by high immunogenicity, capable of inducing elevated titers of broadly neutralizing antibodies (bNAbs) against both seasonal H1 and avian H5 subtypes, thus prompting the development of universal influenza vaccines [89]. Another study designed two protein nanoparticle vaccines that were able to display quadrivalent influenza trimeric HA antigens [90]. These nanoparticles were computationally designed with two-component icosahedrons named I53-dn5. Four trimeric HA antigens derived from influenza A H1, influenza A H3, influenza B/Yam, and influenza B/Vic were co-displayed on nanoparticles and formed the qsMosaic-I53-dn5 vaccines. Simultaneously, four individual HA-displaying nanoparticles were equally mixed to form qsCocktail-I53-dn5 vaccines. Upon vaccination in mice, ferrets, and non-human primates (NHPs), both nanoparticle vaccines induced stronger immune responses compared to commercial quadrivalent influenza vaccines (QIVs). Notably, these nanoparticles induced a broad spectrum of protective antibody responses against heterologous viruses, which were primarily accounted for by nAbs targeting conserved HA stem regions. Therefore, I53-dn5-based multivalent influenza nanoparticle vaccines were anticipated to emerge as supraseasonal influenza vaccine candidates. To mitigate bacterial ferritin-induced immunogenicity, another group utilized self-assembling recombinant human heavy chain ferritin (rHF) as the nanoparticle carriers to present triple ectodomains of influenza matrix protein 2 (3M2e) [91]. The 3M2e-rHF nanoparticle vaccines were intranasally administrated to mice. Both M2e-specific IgG humoral immune responses and T cell immune responses were significantly induced. The non-adjuvanted nanoparticle vaccine also induced a strong mucosal IgA immune response. Most importantly, the 3M2e-rHF nanoparticle vaccines protected mice against both homologous H1N1 and heterologous H9N2 influenza viruses, providing a new strategy for the development of broad-spectrum anti-influenza vaccines. To develop potential universal vaccines against both influenza A and B viruses, the same research group employed the rHF nanoparticle to display the conserved epitopes of the A α-helix of HA, the ectodomain of matrix protein 2, and the HCA-2 of NA, resulting in the generation of HMNF nanoparticle vaccines [92]. After intranasal immunization in mice, robust titers of antigen-specific antibody responses and high levels of cellular immune responses were observed. Remarkably, high levels of antibody titers and T cell ratios were still detected in mice after 180 days post-immunization with this polyvalent vaccine. The HMNF nanoparticle vaccine also induced protective efficacy against diverse subtypes of influenza A and B viruses, highlighting its potential as a promising universal influenza vaccine.

Virus-like particles (VLPs) are formed by the self-assembly of viral structural proteins. VLPs do not contain viral genetic materials and are potentially safe. Che-Ming Jack Hu et al. designed a VLP vaccine composed of HA, NA, and matrix protein (M1) derived from a human isolate A/Taiwan/S02076/2013(H7N9) [93]. This VLP vaccine elicited potent humoral and cellular immune responses in murine and avian animal models, providing options for avian influenza control in human and animal environments. Zhiguang Ren et al. developed similar H7N9 A/Shanghai/2 VLPs containing HA, NA, and M1 using the baculovirus (BV) expression system [94]. The developed VLPs induced strong humoral and cellular immune responses, as well as strong lung IgA and lung tissue-resident memory (TRM) cell-mediated local immune responses. Additionally, VLPs provided complete protection against the fatal H7N9 A/Shanghai/2/2013 virus infection. Plant-derived viruses hold great promise as vectors of VLPs. Jérôme Denis et al. inserted conserved influenza M2e epitopes into the C-terminal of papaya mosaic virus (PapMV) coat proteins (CPs) to engineer influenza vaccines based on PapMV VLPs [95]. The resulting PapMV-CP-M2e VLPs were not only carriers of efficient antigen presentation but also served as adjuvants in vaccine formulations. Compared to alum adjuvants, PapMV-CP VLP adjuvants induced stronger humoral immune responses upon the co-delivery of M2e peptides in vivo. Additionally, PapMV-CP-M2e VLPs provided full protection against the influenza A/WSN/33 virus (H1N1) challenge. VLPs can also be generated utilizing other human pathogenic viruses. Jiangxue Wei et al. developed a biomimetic dual-antigen influenza vaccine based on a hepatitis B virus core VLP (HBc VLP) [96]. The influenza M2e antigens were externally displayed on VLPs, while the influenza nucleoprotein (NP) peptides were encapsulated within these VLPs. In comparison to other single-antigen vaccines, the biomimetic VLP influenza vaccine induced stronger humoral immunity and cellular response in mice, providing protection against the lethal challenge of H1N1 viruses. Given that both M2e and NP proteins were relatively conserved influenza antigens, this dual-antigen VLP vaccine provided a new strategy for the development of an effective universal vaccine. VLP vaccines have attracted more attention due to their high safety and effectiveness. VLP vaccines provide a safe, efficient, and cost-effective method for the development of influenza vaccines and have broad prospects in the field of influenza vaccines.

Inorganic nanoparticles also have been extensively utilized in the development of influenza vaccines. Due to their exceptional biocompatibility, ease of preparation, and scalable productivity, inorganic nanoparticles can be precisely designed into specific shapes and sizes, which have been developed as vaccine adjuvant components [54]. The inorganic gold nanoparticles (AuNPs) are promising vaccine carriers that can be easily absorbed by antigen-presenting cells (APCs), including dendritic cells (DCs) and macrophages [97]. Chao Wang et al. designed a dual-linker AuNP, which multivalently conjugated both influenza A (H3N2) HA trimers and bacterial flagellin (FliC) [98]. The HA proteins are the major antigens of influenza A viruses and can trigger protective immunity by inducing nAbs. FliC serves as a vaccine adjuvant, which is an agonist of toll-like receptor 5 (TLR5) on immune cells. The formulated AuNPs-HA/FliC vaccines significantly promoted the uptake of antigens and induced the secretion of cytokines, resulting in enhanced T cell proliferation. M2 is another membrane protein of the influenza virus, which is involved in the processes of viral uncoating, viral assembly, and the release of new virions. The extracellular portion of M2 (M2e) has been utilized as an attractive antigen for influenza vaccines. Wenqian Tao et al. covalently loaded M2e antigens onto AuNP carriers adjuvanted with soluble CpG. The conjugation of M2e-AuNPs also attracted substantial amounts of free M2e molecules around AuNPs. The intranasal immunization of CpG-adjuvanted M2e-AuNPs generated strong M2e-specific antibodies in mice, which provided strong and long-lasting protection against deadly influenza attacks [99]. Studies also have shown that silver nanoparticles (AgNPs) can be utilized as an immunomodulator, which promotes the secretion of pro-inflammatory cytokines [100,101]. Influenza virus-inactivated vaccines with AgNPs as adjuvants significantly enhanced mucosal immunity, which was characterized by the elevated antigen-specific IgA tiers and corresponding plasma cells [102]. Most importantly, these AgNP-adjuvanted vaccines protected mice from lethal influenza infection. AgNPs induced stronger IgA production and lower toxicity compared with other commercial adjuvants.

Apart from the above metallic nanoparticles, non-metallic nanoparticles also have been utilized as potent vaccine adjuvants. Calcium phosphate (CaP) has the advantages of pH-dependent dissolution and stability, making it easier to produce and store. Biocompatible non-metallic calcium phosphate nanoparticles (CaPNPs) have been found to enhance the immune response [103,104]. Additionally, CaP is a component of vertebrate bones and other tissues, which can be well-tolerated and absorbed by the human body, making CaPNPs adjuvant potentially safe. Tulin Morcöl et al. evaluated three different doses of inactivated influenza vaccines with CaPNPs as adjuvants and carriers [105]. The utilization of CaPNPs resulted in a reduction in antigen consumption and induced significantly higher titers of antiviral antibodies in serum than those in the non-adjuvanted group. Th1-typed immune responses are associated with IgG2a stimulation. Vaccines incorporating CaPNP adjuvants exhibited a well-balanced Th1/Th2 type antibody response compared to vaccines adjuvanted with aluminum. In addition, after being challenged with a lethal dose of the 2009 (H1N1pdm) live virus, mice injected with CaPNP-adjuvanted vaccines showed protective effects against viral infection. The silica nanoparticles (SiNPs) have emerged as promising drug delivery carriers due to their high specific surface area, feasible surface functionalization, and favorable biocompatibility. Meanwhile, SiNPs can also enhance the immune response and have the potential to be used as vaccine adjuvants [106,107]. Vanessa Neuhaus et al. developed a double-adjuvanted vaccine against the H1N1 influenza virus utilizing plant-produced H1N1 HA as the antigen and employing SiNPs as well as bis-(3′,5′)-cyclic dimeric guanosine monophosphate (c-di-GMP) as adjuvants [108]. After intratracheal administration of the double-adjuvanted vaccine in mice, the systemic humoral immune response was significantly induced to produce antigen-specific antibodies. Additionally, the SiNP and c-di-GMP double-adjuvanted vaccine also induced strong mucosal immune responses, which were characterized by higher local IgG, IgA, and T cell responses within the bronchoalveolar lavage (BAL). Nanodiamond (ND) is a kind of carbon nanoparticle that has attracted much attention within biomedical fields due to its advantages of chemical inertness, low cost, and non-toxicity. Studies showed that the trimeric H7 antigens of avian influenza A H7N9 virus could be bound onto the surface of ND particles [109]. These ND-adjuvanted antigens elicited stronger humoral immune responses compared to the free trimer H7, which demonstrated that ND nanoparticles can serve as potent adjuvant components for the design of innovative vaccines. Although numerous metallic and non-metallic nanoparticles have been utilized in influenza vaccine development, particularly in adjuvant optimization, none of these nanoparticle-based vaccines have been approved for clinical use. Their efficacy, safety, and biocompatibility should be further evaluated in rigorous clinical trials.

Due to the diversity of influenza subtypes caused by antigen drift and antigen shift, the protection efficacy of traditional influenza vaccines is unsatisfactory. However, the nanoparticle-based influenza vaccine has the advantage of displaying more antigens than the traditional influenza vaccine, thereby enabling broader and more effective immune responses. Consequently, nanoparticle vaccines and adjuvants provide a new approach for the research and development of broad-spectrum anti-influenza vaccines and are expected to be developed into practical vaccines in the future.

**Table 1 vaccines-12-00030-t001:** The applications of nanoparticles as delivery systems or adjuvants in influenza vaccines.

Viruses	Vaccines	Nanoparticles	Sizes	Functions	Status	Ref.
Influenza	CNPs-KAg	Chitosan	571.7 nm	Induce high levels of antigen-specific IgG antibodies and mucosal IgA antibodies	Pre-clinical	[85]
	PLGA-KAg	PLGA	200 nm to 300 nm	Reinforce the expansion of antigen-specific lymphocytes	Pre-clinical	[86]
	PNP hydrogel	PEG-PLA/ HPMC-C_12_	31 nm	Enhance the potency, durability, and breadth of humoral immune responses	Pre-clinical	[87]
	HA-ferritin	Ferritin	30 nm	Induce stronger immune responses than licensed inactivated vaccines	Phase I	[88]
	H2HA-ferritin	Ferritin	37.23 nm	Induce high titers of broadly neutralizing antibodies	Phase I	[89]
	qsMosaic-I53-dn5/qsCocktail-I53-dn5	I53-dn5	50 nm	Induce broad-spectrum nAbs against both autologous and heterologous viral strains	Phase I	[90]
	3M2e-rHF	Human heavy-chain ferritin	15.9 nm	Induce M2e-specific IgG humoral immune responses and T cell immune responses, as well as mucosal IgA immune responses	Pre-clinical	[91]
	HMNF	Human heavy-chain ferritin	18 nm	Induce robust and durable titers of antigen-specific antibody responses, along with sustained levels of cellular immune responses	Pre-clinical	[92]
	H7N9 VLP	HA-, NA-, and M1-based VLP	113.9 nm	Elicit potent humoral and cellular immune responses in murine and avian animal models	Pre-clinical	[93,94]
	PapMV-CP-M2e	PapMV VLP	86 nm	Induce stronger humoral immune responses upon co-administration with M2e peptides	Pre-clinical	[95]
	HBc VLP	M2e- and NP-containing HBc VLP	30 nm	Induce stronger humoral immunity and cellular response, which provide protection against the lethal challenge of viruses	Pre-clinical	[96]
	AuNPs-HA/FliC	Gold nanoparticles (AuNPs)	145 nm	Trigger protective immunity by inducing nAbs and cytokine secretion	Pre-clinical	[98]
	M2e-AuNPs	AuNPs	12 nm	Provide strong and long-lasting protection against influenza attacks	Pre-clinical	[99]
	Inactivated vaccine with AgNPs	Silver nanoparticles (AgNPs)	18 nm	Induce stronger IgA production and lower toxicity	Pre-clinical	[102]
	Inactivated vaccine with CaPNPs	Calcium phosphate nanoparticles (CaPNPs)	450 nm to 500 nm	Induce high titers of antiviral antibodies and balanced T cell response	Pre-clinical	[105]
	HA-SiNP/c-di-GMP	Silica nanoparticles (SiNPs)	100 nm to 500 nm	Induce high local IgG, IgA, and T cell responses within BAL	Pre-clinical	[108]
	H7-ND	Nanodiamond (ND)	58 nm to 580 nm	Elicit stronger humoral immune responses than free H7 trimers	Pre-clinical	[109]

## 3. The Applications of Nanoparticles in Coronavirus Vaccines

The coronavirus family is named after its shape resemblance to the “crown”. Coronavirus infection can cause various degrees of respiratory diseases, ranging in severity from self-limited respiratory diseases to fatal pneumonia. Most coronaviruses are prone to mutations, possess strong immune evasion capabilities, and demonstrate broad host tropism and the potential for cross-species transmission [110,111,112]. Currently, there are seven known coronaviruses that pose threats to human life and health [113]. As we have mentioned previously, three pathogenic human coronaviruses can cause severe acute lung injury (ALI) or acute respiratory distress syndrome (ARDS), which include severe acute respiratory syndrome coronavirus (SARS-CoV), Middle East respiratory syndrome coronavirus (MERS-CoV), and SARS-CoV-2. Another four human coronaviruses including HCoV-OC43, HCoV-NL63, HCoV-229E, and HCoV-HKU1 only lead to mild respiratory symptoms but still can cause severe respiratory illness in the elderly and children. The emerging and recurrent coronaviruses pose huge threats to the global economy and human health. Early diagnosis and prompt prevention measures can effectively mitigate the impact of their outbreaks. Vaccination is the most efficacious approach to prevent and control coronavirus disease. Consequently, various types of emergently used or FDA-approved candidate vaccines against COVID-19 have been developed. Alongside the classic inactivated whole virus vaccine and live attenuated vaccine, newer generations of COVID-19 vaccines, such as mRNA vaccines and viral vector vaccines, are widely used during the SARS-CoV-2 pandemic [114,115]. In addition, nanoparticle-based vaccines carrying homologous or heterologous coronavirus antigens effectively extend and enhance the immunogenicity of vaccines, thereby conferring extensive protection against a variety of coronaviruses and SARS-CoV-2 variants and gradually occupying an important position in the development of COVID-19 vaccines.

The spike (S) glycoproteins of coronaviruses dominate the binding between the virus and host cell receptors, playing crucial roles in mediating virus invasion [116]. Meanwhile, the S protein is the primary inducer of the humoral immune response. Most effective neutralizing antibodies (nAbs) against coronavirus infection target S proteins [117]. Receptor recognition represents a crucial step in viral invasion into the host cell. Studies have shown that SARS-CoV, SARS-CoV-2, and HCoV-NL63 mainly exploit the same cellular receptor named the human angiotensin-converting enzyme 2 (hACE2) [118,119,120]. MERS-CoV mainly utilizes the human dipeptidyl peptidase 4 (hDPP4/hCD26) as its receptor [121]. The cellular receptor of HCoV-229E is identified as the human aminopeptidase N (hAPN) [122]. HCoV-OC43 and HCoV-HKU1 infect cells utilizing the 9-O-acetylated sialic acid (9-O-Ac-Sia) as the receptor [123]. The receptor-binding domain (RBD) of the S protein plays a crucial role in the receptor docking process. RBD has been demonstrated to effectively induce the production of nAbs in various coronaviruses [124,125,126]. The N-terminal domain (NTD) of the S protein can bind to co-receptors and is structurally adjacent to RBD, which also possesses the potential to elicit potent nAbs [127]. Both NTD and RBD belong to the N-terminal S1 subunit of the S protein. The C-terminal S2 subunit of S contains multiple domains that mediate the fusion of virus and cell membranes [118]. The fusion peptide (FP), heptad repeat 1 (HR1), central helix (CH), connector domain (CD), and heptad repeat 2 (HR2) within the S2 subunit are relatively conserved regions, which serve as optimal targets for eliciting broad-spectrum nAbs [128]. These potential target regions are frequently employed as immunogens for subunit vaccines or viral vector vaccines, which can effectively stimulate the immune system to produce nAbs, resulting in the inhibition of viral infection or the prevention of severe illness [48,129]. Therefore, these antigens have also been effectively utilized in the development of coronavirus nanoparticle vaccines.

Lipid nanoparticles (LNPs) typically exhibit a diameter of 80–200 nm, which is similar to the size of most viral particles, including coronavirus virions. The ionizable lipid component of LNPs possesses significant adjuvant properties. Therefore, LNPs do not require additional adjuvants [130]. The combination of microfluidic technology can effectively adjust the proportion of phospholipids, cholesterol, and other components in LNPs and quickly and effectively mix to generate nanoparticles in targeted sizes [131]. In addition, manipulating the LNP component enables antigen-specific targeting activity in vivo [132]. Therefore, the LNP represents an efficacious strategy for delivering mRNA and is commonly used in mRNA vaccine design [133]. Due to the ability of both LNPs and mRNA to trigger innate immune responses, there is no need to add additional adjuvants during formulation [134]. The mRNA-1273 and BNT162b2 vaccines, which were developed by Moderna and BioNTech/Pfizer, respectively, were the first batch of mRNA-LNP vaccines authorized for preventing COVID-19 [135]. Both mRNA-LNP vaccines encoded the full length of SARS-CoV-2 S proteins. Designing vaccines against broad ranges of SARS-CoV-2 variants is important to combat different mutant pandemics. Ke Xu et al. designed an innovative S-based mRNA-LNP vaccine named SYS6006, which covered key mutation sites presented in pandemic strains, including Delta, BA.4, BA.5, and BF.7 [136]. Two-round vaccination with SYS6006 induced nAbs against infections of the SARS-CoV-2 original strain and Delta and Omicron BA.2 variants in mice and NHPs. Both SARS-CoV-2-specific memory B and T cell immune responses were induced by SYS6006, which might account for the prolonged protective immunity against subsequent Omicron variants. The SYS6006 mRNA-LNP vaccine has been approved for emergency use in China. Another group utilized SARS-CoV-2 RBDs as antigens to construct an mRNA-LNP vaccine regardless of S proteins [137]. The resulting RBD-based LNP-encapsulated mRNA vaccine (ARCoV) induced robust nAbs as well as T cell immunity in both mice and NHPs, which provided full protection against authentic SARS-CoV-2 infection. Importantly, in contrast to traditional mRNA-LNP vaccines, ARCoV vaccines were highly stable and could be stored at room temperature for over one week. In addition to traditional mRNA-LNPs expressing S or RBD antigens, the inclusion of other viral protein-encoding mRNAs within LNPs has shown enhanced protection efficacy. Notably, a study revealed that dual-antigen mRNA-LNP vaccines encoding both SARS-CoV-2 S and nucleocapsid (N) antigens induced stronger protection against both Delta and Omicron variants compared to mRNA-LNP vaccines expressing S or N alone [138]. The in vivo CD8^+^ T cell depletion assay indicated that the broad protection of mRNA-S+N LNP might benefit from N-specific immunity. Linear mRNAs are susceptible to degradation by the safeguard of cellular innate immunity. Therefore, Liang Qu et al. designed a highly stable circRNA-LNP vaccine that encoded SARS-CoV-2 trimeric RBD antigens [139]. These circRNA-LNPs produced higher amounts and more persistent antigens than regular mRNA vaccines, which also induced high levels of nAbs and Th1-biased immune responses in vivo. CircRNA-LNPs provided strong protection against authentic SARS-CoV-2 infection in both mice and NHPs. Importantly, circRNA-LNPs expressing Delta RBD antigens provided protection against both Delta and Omicron strains, rendering them favorable choices for combating current SARS-CoV-2 variants. In clinical trials, mRNA-LNP vaccines demonstrated more than 90% effectiveness against the original SARS-CoV-2 strain and achieved long-lasting immune protection [140,141]. However, with the emergence of various variants of SARS-CoV-2, the inhibitory efficacy of LNP vaccines on variants of concern (VOCs) was significantly reduced, and they failed to exert potent protection effects [142]. In addition, the storage conditions for mRNA-LNP vaccines are relatively strict, requiring cold chain distribution and storage, which limits their applicability in less developed regions [143]. Thus, searching for alternative storage conditions is crucial for advancing mRNA-LNP development.

Subunit proteins based on viral RBDs are often limited by their small size and stable spatial conformation, which hampers their immunogenicity. Increasing the size or quantity of antigens through tandem expression or parallel display can improve the immune efficacy of the vaccine [48]. Self-assembling nanoparticle proteins can simultaneously display multiple different types of antigenic epitopes on the surface, demonstrating their advantages in preparing antiviral multivalent vaccines [89]. The SpyCatcher/SpyTag system is currently a commonly used “plug and play” platform for designing self-assembling nanoparticle vaccines [144]. By fusing the SpyCatcher domain with subunits of nanoparticle proteins, the SpyTag-labeled antigens can be covalently conjugated to nanoparticles via an isopeptide bond between SpyTag and SpyCatcher. Other analogous docking platforms include the DogCatcher/DogTag system and the SnoopCatcher/SnoopTag system. Vaccines prepared utilizing these technologies can exert strong antiviral and protective effects in animal models [145,146]. These protein coupling systems provide strong support for the synthesis of protein nanoparticle vaccines. Commonly used self-assembling nanoparticle frameworks include 24-polymer ferritin, 60-polymer mi3, 60-polymer E2p, 60-polymer lumazine synthase (LS), and 60-polymer bicomponent protein nanoparticles (I53-50) [145,147,148,149]. The first SARS-CoV-2 self-assembling nanoparticle vaccine was designed by displaying 60 copies of SARS-CoV-2 RBD proteins on the two-component protein nanoparticle I53-50 (Table 2) [150]. RBD-I53-50 nanoparticles induced an nAb titer approximately 10 times higher than that induced by the S protein at an immune dose of 1/5 in mice, demonstrating a robust humoral immune response against SARS-CoV-2. Our group previously designed a dual-antigen nanoparticle vaccine by covalently coupling the RBD and HR regions of a SARS-CoV-2 S protein with self-assembled 24-mer *H. pylori* ferritin through the SpyCatcher/SpyTag system [147]. This RBD/HR ferritin nanoparticle vaccine induced strong nAbs against authentic SARS-CoV-2. Additionally, it also showed potential cross-neutralizing activities against other coronaviruses. Both antigen-specific B cell and T cell immune responses in mice and rhesus monkeys were significantly higher than those in the RBD/HR monomer groups. Another group designed a SpyCatcher003-mi3 protein nanoparticle platform, which was engineered from aldolase of thermophilic bacteria [145]. Approximately 56 copies of SARS-CoV-2 RBDs were conjugated to the mi3 nanoparticle via SpyCatcher003/SpyTag003. The resulting nanoparticle vaccines induced strong nAbs responses in mice and pigs, which were higher than those in convalescent human sera. Importantly, the RBD-mi3 nanoparticles were thermostable and could be lyophilized without compromising immunogenicity, which facilitated easier global distribution. The conjugation of dimer or trimer antigens to nanoparticles can display more antigens. Qibin Geng et al. successfully conjugated Fc-tagged SARS-CoV-2 RBD dimers to 60-mer lumazine synthase (LS), generating the 120-mer RBD nanoparticle vaccine [55]. The resulting RBD nanoparticle vaccines elicited potent nAbs against various SARS-CoV-2 mutants, SARS-CoV, and SARS-CoV-related bat coronaviruses. These nAbs persisted for at least two months. In addition to the RBD antigen, one group utilized the C-terminal truncated S ectodomain (S∆C) as the antigen. The S∆C proteins were conjugated to the ferritin nanoparticle, which constituted the DCFHP nanoparticle vaccine [151]. The formulation of DCFHP with an aluminum hydroxide (alum) adjuvant could elicit potent, durable, and broad-spectrum neutralizing antibodies against nearly all the VOCs. Furthermore, the DCFHP-alum could maintain strong immunogenicity for more than two weeks at temperatures exceeding room temperature. Another group utilized the 60-mer lumazine synthase (LS) as the inner core, and conjugated S trimers of SARS-CoV-2 onto the 60-mer nanoparticle, forming the SARS-CoV-2 S-LS nanoparticle vaccine [152]. These nanoparticle vaccine-elicited nAbs were 25-fold higher than those induced by the S-only vaccine, which indicated that the LS-based nanoparticle vaccine could induce stronger humoral immune responses. In order to achieve the effectiveness of vaccines against different coronaviruses, even animal-derived betacoronaviruses, scientists utilized the mi3 nanoparticle to construct a “mosaic” nanoparticle vaccine, which simultaneously displayed 60 randomly arranged RBDs derived from eight betacoronaviruses, including SARS-CoV-2 [153,154]. Compared to nanoparticles expressing SARS-CoV-2 RBD alone, these mosaic RBD nanoparticles exhibited stronger heterologous cross-reactive binding and neutralizing characteristics and could protect animals from attacks from multiple betacoronaviruses. More importantly, the vaccination of mosaic RBD nanoparticle vaccines had the potential to simultaneously protect against future betacoronavirus spillovers. The effectiveness of self-assembling vaccines relies on their conjugation strategies, the uniformity of nanoparticles, nanoparticle stability, and the manufacturing techniques employed. Therefore, choosing an appropriate and effective conjugation system is paramount. Meanwhile, the proper antigen design will also contribute to the improvement of nanoparticle vaccine effectivity.

Virus-like particles (VLPs) are formed by the self-assembly of viral structural proteins in vitro, mimicking the structure and characteristics of natural viral particles without the introduction of viral genetic materials [155]. VLPs can directly pass through lymph nodes, thereby promoting antigen presentation and inducing effective T and B cell responses [156]. Ki-Back Chu et al. constructed S, S1, or S2-containing VLPs utilizing Sf9 cells (Table 2) [157]. The influenza M1 proteins were treated as the inner core proteins. Mice immunization experiments indicated that only full-length S- or S1-containing VLPs were able to induce nAbs, highlighting the significant immunogenicity of S1 subunits. Another group utilized HEK293 cells to produce the VLP vaccine, which contained four structural proteins (S, M, N, and E) of SARS-CoV-2 [158]. The four-component VLPs triggered high titers of IgG against S, RBD, and N in mice, rats, and ferrets, leading to multifunctional Th1-biased T cell responses, reduced viral load, and mitigated lung lesions. It should be noted that VLPs derived from mammalian cells may contain residue immunogenic host proteins. To address this issue, one group produced S trimer-displaying VLPs (CoVLPs) utilizing plant cells, the lipid envelope of which was derived from plant cell plasma membranes [159]. The Phase I clinical trial of this plant-derived VLP showed that CoVLPs were well tolerated and could induce nAbs titers over 10-fold higher than those from COVID-19 convalescent sera. The form of SARS-CoV-2 S protein may influence its stability and immunogenicity. Thus, one study constructed an enveloped VLP (eVLP) that expressed a modified prefusion form of S [160]. The eVLPs were produced utilizing murine leukemia virus (MLV)-based eVLPs, which were referred to as MLV-Gag eVLPs. The prefusion S protein was fused with a transmembrane cytoplasmic terminal domain (TMCTD) of VSV-G, which was referred to as SPG. The resulting SPG-eVLPs were able to induce robust and sustained nAbs, exceeding those from COVID-19 convalescent sera. Additionally, the single immunization of SPG-eVLPs was capable of providing high efficacious protection against authentic SARS-CoV-2 infection in hamster animal models. All aforementioned VLPs relied on the in vitro assembly of target proteins. Recently, a study successfully designed an eVLP that was capable of self-assembling within host cells [161]. An ESCRT- and ALIX-binding region (EABR) was connected to the cytoplasmic region of SARS-CoV-2 S, which recruited ESCRT proteins to induce the budding of eVLPs from host cells. To achieve the in vivo assembly of S-EABR eVLPs, mRNA-LNPs, which were encoded in S-EABR, were delivered into mice. Consequently, S-EABR elicited nAb responses in the form of both membrane anchors and eVLPs. The EABR-based VLP construction technology not only enhanced the potency of elicited nAbs but also extended the breadth of induced immune responses, which enabled the long-term protection against SARS-CoV-2 virus as well as corresponding mutants. VLPs are highly promising vaccines, and the effectiveness evaluation of these vaccines is predominantly conducted in animal models. Further clinical trials need to be conducted to evaluate their safety and efficacy.

The design of polymer-based nanoparticle vaccines targeting SARS-CoV-2 heavily relies on synthetic polymers, such as polylactic acid hydroxyacetic acid (PLGA) and polysorbate 80 (PS80), as well as natural polymers, such as chitosan and saponin. The COVID-19 nanoparticle vaccine NVX-CoV2373 was developed based on the full-length S protein, including transmembrane domain (TM) and cytoplasmic tail region (CT), the stable prefusion conformation of which was confirmed by cryo-election microscopy [162,163]. The Sf9 cell-expressed S trimers were assembled with the PS80 core to form the nanoparticle vaccines, which were co-formulated with the saponin-based Matrix-M adjuvant. The composite vaccines elicited more multifunctional T cell and B cell immune responses than S trimers only and showed significant effects in preventing SARS-CoV-2 infection. Multiple clinical trials have fully confirmed their safety and effectiveness in adults and adolescents [164,165,166]. The NVX-CoV2373 nanoparticle vaccine has been authorized for emergency use in numerous countries. PLGA material exhibits high biocompatibility with the property of favorable degradation under human physiological conditions, rendering it an ideal nanomaterial for drug delivery [167]. Studies have shown that the recombinant SARS-CoV-2 S1 and E (rS1-E) bivalent antigens, which were coated on the surface of PLGA, were able to induce high nAbs titers and enhanced cellular immune responses, which indicated that the rS1-E-PLGA nanoparticle vaccines might be severed as a potential booster for vaccination [168]. Aluminum hydroxide microgels (alums) represent the most widely used licensed adjuvants. Sha Peng et al. developed a particulate alum via a Pickering emulsion (PAPE) nanoparticle adjuvant [169]. Upon co-administration with SARS-CoV-2 RBD antigens, PAPE-RBD elicited 6-fold higher antigen-specific antibody titers and 3-fold more IFN-γ-secreting T cells than conventional alum-RBD-treated mice, indicating enhanced humoral and cellular immune responses. The PAPE adjuvant was packed on the squalene/water interphase utilizing the single-step sonication. The resulting PAPE droplets were able to absorb more RBD antigens and demonstrate higher affinities for DC uptake. The PAPE forms of alums significantly enhanced the efficacy of conventional alums. Similar to influenza vaccines, AuNPs also have been found to enhance the immunogenicity of coronavirus vaccines. One study showed that AuNP-conjugated SARS-CoV-2 RBD antigens were able to induce stronger long-term humoral responses than monomers [170]. However, another study showed that AuNP-adjuvanted SARS-CoV S antigens failed to elicit protective nAbs and mitigate eosinophilic infiltrations in lungs, although AuNP-S vaccines induced higher antigen-specific IgG titers [171]. Thus, more animal experiments need to be conducted to elucidate the exact effectivity and function of AuNP-adjuvanted vaccines on different coronavirus infections. Both organic and inorganic nanoparticles possess distinct advantages as adjuvants or carriers in vaccines. One group designed a composite nano-carrier coalescing both the organic chitosan and the inorganic gold nanostar (AuNS), which exhibited a significant synergistic effect on the efficacy of DNA vaccines [172]. They utilized AuNS-chitosan to intranasally deliver a DNA vaccine encoding SARS-CoV-2 S proteins in mice. The AuNS-chitosan-conjugated DNA vaccine induced strong mucosal immune responses, including elevated levels of S-specific mucosal IgA and lung tissue-resident memory T (TRM) cells, inducing potent systemic humoral immune responses, which were represented by high levels of IgG nAbs. These activated immune responses provided durable protection against both the original SARS-CoV-2 and subsequent variants. Polymer-based nanoparticles can achieve targeted delivery of antigens in vivo based on the advantage of nanoscale size by coating or encapsulating antigens. Further NHP experiments and clinical trials on these nanoparticle-adjuvanted or delivered vaccines need to be conducted to fully confirm their efficacy and safety.

Although numerous nanoparticles have been exploited to develop carriers or adjuvants of novel coronavirus vaccines, few have been successfully approved for clinical use. Their efficacy and safety need to be carefully evaluated in NHP models and clinical trials. Additionally, the stability, biocompatibility, and uniformity of nanoparticles need to be further enhanced or optimized during manufacturing processes. Given that nanoparticle carriers and adjuvants reveal superior advantages in humoral, mucosal, and cellular immune responses compared to traditional delivery systems and adjuvants, nanoparticles merit being applied to more types of different coronaviruse vaccines.

**Table 2 vaccines-12-00030-t002:** The applications of nanoparticles as delivery systems or adjuvants in coronavirus vaccines.

Viruses	Vaccines	Nanoparticles	Size	Functions	Status	Ref.
Coronavirus	mRNA-1273	LNP	80 nm to 150 nm	Elicit potent neutralizing antibodies and mitigate severe illness	Approved (Moderna)	[135]
	BNT162b2	LNP	80 nm to 150 nm	Elicit a high nAbs titer and provide efficient preventive protection	Approved (Pfizer-BioNTech)	[135]
	SYS6006	LNP	80 nm to 150 nm	Induce nAbs against SARS-CoV-2 VOCs in vivo and SARS-CoV-2-specific memory B and T cell immune responses	Approved (CSPC)	[136]
	ARCoV	LNP	88.85 nm	Induce strong nAbs and T cell immunity with high stability	Approved (Walvax)	[137]
	mRNA-S+N	LNP	80 nm to 150 nm	Elicit enhanced protection against both Delta and Omicron variants	Pre-clinical	[138]
	circRNA-RBD	LNP	100 nm	Generate higher amounts and more persistent antigens and induce elevated levels of nAbs and Th1-biased immune responses	Pre-clinical	[139]
	RBD-I53-50	I53-50	37 nm to 41 nm	Induce high nAb titers with low antigen doses	Approved (SK Bioscience)	[150]
	RBD/HR-ferritin	Ferritin	15 nm to 20 nm	Induce strong nAbs with potential cross-neutralizing activity	Pre-clinical	[147]
	RBD-mi3	mi3	20.7 nm	Thermostable and induce stronger nAbs responses than those in convalescent sera	Pre-clinical	[145]
	RBD-Fc-LS	Lumazine synthase (LS)	15 nm	Elicit potent and persistent nAbs against SARS-CoV-2 mutants, SARS-CoV, and other bat coronaviruses	Pre-clinical	[55]
	DCFHP	Ferritin	15 nm to 30 nm	Elicit potent, durable, and broad-spectrum neutralizing antibodies against nearly all the VOCs	Pre-clinical	[151]
	S-LS	LS	63.2 nm	Elicit 25-fold higher nAbs than S only	Pre-clinical	[152]
	Mosaic RBD-mi3	mi3	33 nm to 48 nm	Exhibit strong cross-reactive binding and neutralizing activities and prevent multiple betacoronaviruses attacks	Pre-clinical	[153,154]
	S/S1/S2 VLP	M1-based VLP	80 nm to 200 nm	S- or S1-containing VLPs induce nAbs	Pre-clinical	[157]
	S/M/N/E VLP	VLP	117 nm to 127 nm	Trigger high titers of IgG, leading to multifunctional Th1-biased T cell responses	Phase II	[158]
	CoVLP	Plant-derived VLP	80 nm to 120 nm	Well tolerated and induces 10-fold higher nAbs than convalescent sera	Phase II/III	[159]
	SPG-eVLP	MLV Gag-based VLP	100 nm to 140 nm	Induce robust and sustained nAbs and protect against authentic viruses	Phase I/II	[160]
	S-EABR eVLP	ESCRT-based VLP	40 nm to 60 nm	Achieve in vivo assembly and provide long-term protection against viral mutants	Pre-clinical	[161]
	NVX-CoV2373	PS80/Matrix-M	27.2 nm	Elicit multifunctional T cell and B cell immune responses	Approved (Novavax)	[162,163,164,165,166]
	rS1-E-PLGA	PLGA	670 nm	Induce high nAbs titers and enhance cellular immune responses	Pre-clinical	[168]
	PAPE-RBD	PAPE	3551 nm	Induce higher antigen-specific nAb titers and more IFN-γ-secreting T cells than conventional alum-RBD vaccines	Pre-clinical	[169]
	RBD-AuNP	AuNP	50 nm to 60 nm	Induce stronger long-term humoral responses than monomers	Pre-clinical	[170]
	S DNA vaccine with AuNS-chitosan	Gold nanostar (AuNS)/Chitosan	35 nm to 48 nm	Induce both strong mucosal immunity and potent systemic humoral immune responses	Pre-clinical	[172]

## 4. The Applications of Nanoparticles in HIV Vaccines

Human immunodeficiency virus (HIV) is globally prevalent and is the causative virus of acquired immune deficiency syndrome (AIDS). HIV/AIDS represents a significant global public health issue. It has claimed 40.4 million human lives by the end of 2022. It has been estimated that 85.6 million individuals have been infected with HIV since the beginning of the pandemic. HIV exists in two major types, namely HIV-1 and HIV-2 [16]. HIV-1 is the predominant type and is widespread globally, while HIV-2 has lower virulence and is mainly prevalent in West Africa. Currently, the most effective approach to treating AIDS is performing combined antiretroviral therapy (cART), which involves administrating multiple antiretroviral drugs to suppress viral replication and prevent the frequent emergence of drug-resistant strains [173,174,175]. However, cART only targets actively replicating viruses and has little impact on the reservoir of latent viruses, which is the major obstacle to eradicating HIV [176]. Despite its effectiveness in controlling HIV infection and prolonging patients’ lives, ART cannot cure AIDS. Therefore, there is an urgent need to develop vaccines that can effectively prevent HIV infection. Unfortunately, all candidate vaccines that have undergone clinical trials have either failed or demonstrated limited efficacy in the prevention of HIV infection. Early attempts at developing HIV vaccines were based on inactivated virus methods, immune responses of which were stimulated using inactivated viruses or their components [177]. However, these vaccines have not been fully evaluated in Phase II/III clinical trials. Other attempts using soluble subunit protein vaccines also failed to elicit protective responses against HIV infection [178,179]. These vaccines were designed to activate the immune system using components like HIV-1 gp120, which is the target protein for inducing neutralizing antibodies (nAbs) and the main focus for vaccine design. The gp120 protein is located on the outer surface of virions and plays a role in recognizing host cells and binding to the CD4 surface receptor on these cells [180]. On the other hand, the transmembrane protein gp41 plays a crucial role in the fusion process between the viral membrane and the cell membrane [181]. The extensive variability of HIV and its immune evasion mechanisms have hindered the success of gp120 and gp41-based vaccines in clinical trials. Notably, the RV144 vaccine Phase III trial conducted in Thailand in 2009 demonstrated a protection efficacy of 31.2%. The RV144 vaccine was a combination of recombinant canarypox vector vaccine (ALVAC-HIV (vCP1521)) and recombinant gp120 subunit vaccine (AIDSVAX B/E) [182]. It is currently the only vaccine that has demonstrated efficacy against HIV infection.

The development of nanotechnology has brought new hope to HIV vaccine research. Self-assembling protein nanoparticle vaccines have been extensively studied for preventing HIV infection, which include 24-mer ferritin, 60-mer lumazine synthase (LS), 60-mer E2p, and many others (Table 3). Bacterial ferritin proteins can self-assemble into extremely stable 24 polymers, with the N-terminal facing outside the nanoparticles. Theoretically, these nanoparticles can present 24 copies of antigens on their surface. Widely used ferritin nanoparticles derive from *Helicobacter pylori* (*H. pylori*) and *Pyrococcus furiosus* (*P. furiosus*). Kwinten Sliepen et al. utilized *H. pylori* ferritin to display 24 copies of native-like HIV envelope glycoprotein BG505 SOSIP.664 gp140 trimers [183]. These trimer-conjugated nanoparticle vaccines significantly enhanced the immunogenicity of gp140 in mice and rabbits and induced more nAbs against most tier 1A viruses, several titer 1B viruses, and the autologous titer 2 virus. Other *H. pylori* ferritin-based nanoparticle vaccines were designed to display HIV CH848 10.17DT SOSIP trimers [184]. The conjugation of nanoparticles significantly upregulated germinal center (GC) B cells and Tfh cells. Nanoparticle-elicited nAbs could neutralize both autologous and heterologous viruses, revealing the capability of initiating V3-glycan broadly neutralizing antibody (bNAb) B cell lineages. Similarly, Talar Tokatlian et al. designed an MD39 gp140-conjugated *P. furiosus* ferritin nanoparticle vaccine, which formed eight copies of gp140 trimers (MD39-8mer) on the surface of nanoparticles [185]. MD39-8mer nanoparticle vaccines were able to quickly penetrate GCs and be captured by follicular dendritic cells (FDCs). Typically, the “glycan shield” of envelope proteins is thought to hinder the production of nAbs. However, MD39-8mer vaccination experiments indicated that the efficient antigen presentation and nAbs induction were complement-, mannose-binding lectin (MBL)-, and antigen glycan-dependent. All the above ferritin nanoparticles are assembled in vitro and are purified from cultured cells. One group successfully constructed an mRNA-LNP vaccine to assemble *H. pylori* ferritin nanoparticles in vivo [186]. The mRNA-LNP, which encoded CH848 10.17DT SOSIP trimer-ferritin nanoparticles, was immunized in bNAb precursor VH + VL knock-in mice. These in vivo assembled nanoparticles initiated the expansion of bNAb precursor B cells and induced bNAbs against both autologous tier 2 viruses and heterologous HIV isolates. In addition to 24-mer nanoparticles, scientists also discovered or designed many 60-mer nanoparticles that were capable of presenting 60 copies of antigens. Joseph Jardine et al. utilized 60-mer lumazine synthases (LSs) derived from the hyperthermophile *Aquifex aeolicus* to present HIV antigens [149]. They designed several germline-targeting (GT) variants of engineered HIV gp120 outer domain (eOD) immunogens and identified eOD-GT6, which bound to GL VRC01-class nAbs with high affinities. The resulting eOD-GT6 LS nanoparticle vaccine potently activated both germline (GL) and mature B cells, providing compelling evidence that germline-targeting strategies could help induce potent bNAbs. Designing vaccines targeting conserved regions is another strategy to induce bNAbs. The membrane-proximal external region (MPER) of the HIV gp41 ectodomain is one of the most conserved regions of Env and elicits three well-studied bNAbs, including 2F5, 4E10, and 10E8 [187]. A study was conducted to conjugate MPER to the E2 protein (E2p) of *Geobacillus stearothermophilus*, which formed self-assembling 60-mer nanoparticles [188]. MPER-E2p elicited strong MPER-specific antibodies, as well as bNAbs, targeting titer 1 and titer 2 viruses upon co-vaccinated with gp160 DNA vaccines. Utilizing the same 60-mer nanoparticle, another group successfully conjugated stabilized HIV gp140 to E2p, displaying 20 copies of gp140 trimers [189]. The gp140-E2p induced more robust stimulation of cognate bNAb VRC01 receptor-containing B cells than gp140 trimers. Similarly, Yi-Nan Zhang et al. displayed 20 copies of HIV BG505 uncleaved prefusion-optimized (UFO) gp140 trimers on E2p nanoparticles, which preserved the native-like Env trimer structure [190]. BG505 UFO gp140-E2p exhibited 420-fold longer retention in lymph node follicles and 20-fold greater presentation on FDC dendrites, as well as induced 4-fold stronger GC reactions than trimer-only vaccines.

Nanoparticles are also widely used as carriers or adjuvants in the field of HIV vaccination. Polylactate-glycolic acid (PLGA), polylactic acid (PLA), and polycaprolactone (PCL) are the most commonly used biodegradable polymer nanoparticles for vaccine delivery (Table 3) [191]. Hajar Rostamia et al. chemically conjugated HIV P24-Nef peptide to FLiC (a flagellin molecule sequence from Pseudomonas aeruginosa) and used PLGA as a carrier to construct the HIV P24-Nef/FLiC nanoparticle vaccine [192]. FLiC was treated as a TLR5 agonist, while PLGA was used as an efficient vaccine transmitter to immune systems. The HIV P24-Nef/FLiC/PLGA nanoparticle vaccine enhanced the immunogenicity of the antigen and reduced the required antigenic dose, resulting in heightened cellular immune responses. Another study conjugated poly (maleic anhydride-ALT-1-octadecene) (PMHC_18_) with poly (ethylene glycol) (PEG) to synthesize the amphiphilic polymer (P_1_M_10_) nano-carriers for HIV vaccines [193]. The HIV Env nanoparticles (Env/NPs) produced utilizing P_1_M_10_ induced more potent and broader neutralizing antibodies against various HIV subtypes. Importantly, the resulting Env/NP vaccines were stable under different storage conditions. Apart from organic nanoparticles, inorganic nanoparticles are also used in HIV research. These inorganic nanoparticles have direct antiviral effects. For example, Jose Luis Elechiguerra et al. demonstrated that silver nanoparticles (AgNPs) inhibited the binding between HIV viruses and host cells [194]. The inhibition was accomplished by the direct binding of AgNPs to HIV gp120 glycoproteins. Humberto H. Lara et al. found that adding AgNPs to nAbs significantly improved their neutralizing potency against cell-associated HIV infection, although the underlying mechanisms require further research [195]. Inorganic nanoparticles can also be used as vaccine carriers or adjuvants to enhance vaccine bioavailability and immune responses [196,197]. Gold nanoparticles (AuNPs) have been extensively investigated for their potential in HIV vaccine research. Surface-engineered gold nanorods (AuNRs) have been reported by Ligeng Xu et al. to serve as promising adjuvants for DNA vaccines targeting HIV [196]. These modified AuNRs significantly enhanced cellular and humoral immune responses. Compared with the HIV Env-expressing DNA-only vaccine, the AuNR-conjugated HIV DNA vaccines could activate APCs better, resulting in enhanced antigen presentation. Núria Climent et al. found that AuNPs loaded with dendritic cells carrying HIV peptides and mannosides enhanced HIV-specific CD4^+^ and CD8^+^ T cell proliferation and induced high levels of cytokine secretion, which indicated enhanced HIV-specific cellular immune responses [197]. Additionally, silica and calcium phosphate nanoparticles (SiNPs and CaPNPs) have been demonstrated as effective delivery systems for HIV vaccines [198,199]. These studies have revealed that compared to CaPNPs coupled with Env trimers using a random orientation approach, CaPNPs displaying orthogonally arranged Env trimers on the surface demonstrated superiority in activating Env-specific B cells and inducing Env-specific antibody responses [198]. These results indicated that the covalent coupling of HIV-1 Env native-like trimers to CaPNPs could maintain protein conformation better. Shuang Li et al. demonstrated that an HIV Env trimer-expressing DNA vaccine, packaged within CpG-functionalized silica-coated CaPNPs (SCPs), elicited broad humoral and robust cellular immune responses in mice and guinea pigs compared to DNA-only vaccines [199]. All of these reports indicate that nanoparticles can serve as excellent HIV vaccine carriers or adjuvants, which safeguard the effectiveness of vaccines.

Until now, no effective HIV vaccine is available for high-risk populations. A therapeutic HIV vaccine is also unreachable. Because of the high HIV variability and immune evasion, HIV vaccines can hardly induce bNAbs against heterogeneous viruses and corresponding mutants. However, HIV nanoparticle vaccines, especially self-assembling protein nanoparticle vaccines, can simultaneously present 20 to 60 different types of HIV immunogens. These mosaic nanoparticle vaccines not only induce diversified nAbs against corresponding HIV viruses but also induce bNAbs, which can target different HIV mutants. Further clinical trials on HIV nanoparticle vaccines are needed to fully validate their safety and effectiveness.

**Table 3 vaccines-12-00030-t003:** The applications of nanoparticles as delivery systems or adjuvants in HIV vaccines.

Viruses	Vaccines	Nanoparticles	Size	Functions	Status	Ref.
HIV	BG505 SOSIP.664 gp140-ferritin	Ferritin	30 nm to 40 nm	Enhance the immunogenicity of gp140	Pre-clinical	[183]
	CH848 10.17DT SOSIP-ferritin	Ferritin	30 nm to 50 nm	Upregulate GC B cells and Tfh cells	Pre-clinical	[184,186]
	MD39 gp140-ferritin	Ferritin	40 nm	Quickly enter into GCs and be captured by FDCs	Pre-clinical	[185]
	eOD-GT6 LS	LS	32 nm	Activate both germline (GL) and mature B cells and elicit potent bNAbs	Phase I	[149]
	MPER-E2p	E2p	24 nm	Elicit strong MPER-specific antibodies and bNAbs	Pre-clinical	[188]
	gp140-E2p	E2p	37.6 nm	Induce a robust stimulation of cognate bNAb VRC01 receptor-containing B cells	Pre-clinical	[189]
	BG505 UFO gp140-E2p	E2p	41.7 nm	Exhibit long retention in lymph node follicles and a great presentation on FDC dendrites	Pre-clinical	[190]
	P24-Nef/FLiC	PLGA	198 nm	Enhance the immunogenicity of antigens and reduce required antigenic doses	Pre-clinical	[192]
	Env/NP	P_1_M_10_	20 nm	They have a high stability and induce more potent and broader neutralizing antibodies	Pre-clinical	[193]
	Env DNA vaccine with AuNR	Gold nanorod (AuNR)	15 nm	Activate APCs better and enhance cellular and humoral immune responses	Pre-clinical	[196]
	Peptide vaccine with AuNP	AuNP	2.3 nm	Enhance HIV-specific cellular immune responses	Pre-clinical	[197]
	Env-CaPNP	CaPNP	44 nm to 120 nm	Activate Env-specific B cells and induce Env-specific antibody responses	Pre-clinical	[198]
	Env DNA vaccine with SCP	Silica-coated CaPNP (SCP)	158.8 nm	Induce broad humoral and robust cellular immune responses	Pre-clinical	[199]

## 5. The Applications of Nanoparticles in Hepatitis Virus Vaccines

Hepatitis B and hepatitis C are caused by the hepatitis B virus (HBV) and HCV, respectively. Both infections can lead to significant liver damage, including liver cirrhosis and hepatocellular carcinoma. Fortunately, HBV vaccines can provide nearly 100% protection against HBV when given soon after birth [19]. For hepatitis C, direct-acting antiviral medicines (DAAs) can cure more than 95% of HCV-infected individuals, although no effective vaccine has been developed to prevent HCV infection [21,200]. In addition to HBV and HCV, there are also other types of hepatitis viruses such as the hepatitis A virus (HAV), hepatitis D virus (HDV), and hepatitis E virus (HEV). The global transmission of these viruses is relatively less common.

HBV, belonging to the *Hepadnaviridae* family of enveloped viruses, is a double-stranded DNA virus with a genome size of 3.2 kilobases (kbs) [201]. In the field of hepatitis B, vaccines have been widely applied and proven to be highly effective. Since the first release of the hepatitis B vaccine in 1981, it has emerged as one of the primary strategies to prevent hepatitis B [202]. However, due to the limited availability and high costs associated with plasma-derived vaccines, second-generation recombinant hepatitis B virus vaccines were developed and gradually replaced the first-generation ones [203]. Recombinant HBV vaccines were formulated from hepatitis B surface antigen (HBsAg), which provided long-term and HBsAg-specific antibody titers. Compared to traditional vaccines, nanoparticle vaccines possess numerous unique advantages over traditional vaccines, including greater stability and stronger immunogenicity (Table 4) [204]. Saeed Mobini et al. proposed a virus-like particle (VLP)-based anti-HBV vaccine design, placing the antibody-binding epitope of HBsAg on the major immunodominant region (MIR) site of HBcAg to stimulate multilateral immunity. Modeling and molecular dynamics (MD) demonstrated the folding stability of HBcAg as a carrier when inserting Myrcludex and HBsAg into the “a” determinant cluster. The resulting construct is expected to induce both humoral and cellular immune responses against HBV [205]. Currently, the mainstream approaches of hepatitis B vaccines are still the second-generation recombinant vaccines [206]. Despite their widespread use, a small proportion of vaccinated individuals fail to generate sufficient protective antibodies. To address this issue, a third-generation VLP vaccine was developed, which showed better responses in older individuals, obese patients, and those with compromised immune function [207]. These VLP vaccines were produced from Chinese hamster ovary (CHO) mammalian cells and formulated with triple HBV antigens, including HBsAg-S, pre-S1, and pre-S2 [208,209]. Triple-antigen VLP vaccines induced higher rates of seroprotection against HBV with higher nAb levels at lower antigen doses than traditional single-antigen vaccines. In addition to virus-like nanoparticles, self-assembling protein nanoparticles have also gained attention for their unique structural properties and applications in anti-HBV vaccines. Wenjun Wang et al. designed a ferritin nanoparticle vaccine that covalently conjugated 24 copies of HBV pre-S1 to the surface of *P. furiosus* ferritin nanoparticles [210]. The resulting nanoparticle vaccine was capable of delivering antigens to both SIGNR1^+^ dendritic cells and lymphatic sinus-associated SIGNR1^+^ macrophages, thereby activating T follicular helper cells and B cells, respectively. These pre-S1 ferritin nanoparticle vaccines elicited high levels of preventive and therapeutic nAbs, providing next-generation vaccination strategy for the functional cure of hepatitis B. Polymer nanoparticles possess efficient antigen presentation capabilities and good immunostimulatory properties, rendering them increasingly used in the development of anti-HBV vaccines. To improve cellular immune response, Jiahuan Zhu et al. synthesized mannosylated PLGA and prepared mannose-modified nanoparticles (MNPs) loaded with hepatitis B surface antigen (HBsAg) protein. In a mouse model, this PLGA-based polymer nanoparticle vaccine induced persistent humoral immunity and enhanced cellular immune response [211]. Despite the advancements of preventive vaccines against HBV, an effective therapeutic vaccine remains elusive. The development of an effective therapeutic vaccine has been an urgent goal in the medical field. Carrie S.W. Chong et al. loaded hepatitis B core antigen (HBcAg) into PLGA via monophospholipid A (MPLA) and designed a therapeutic hepatitis B vaccine that induced stronger Th1-biased cellular immune responses with more IFN-γ production [212]. Nanoparticle delivery systems provided new avenues for the development of potential therapeutic vaccines.

HCV, a member of the *Flaviviridae* family, is an RNA virus with a single-stranded positive-sense genome. It exhibits a pronounced propensity for chronic infection, which can lead to liver cirrhosis and hepatocellular carcinoma [20]. Currently, there is no effective vaccine for preventing HCV infection. Direct-acting antiviral agents (DAAs) devoid of interferon are the first-line treatment for chronic HCV infection [213]. However, DAAs have limited coverage for global HCV prevalence, as about 80% of chronic infections remain undiagnosed and viral transmission continues. Therefore, there is an urgent need to develop an effective preventive or therapeutic vaccine against HCV [214]. Nanoparticle vaccines for preventing HCV infection have been extensively investigated, including pre-clinical research based on HCV VLP vaccines (Table 4) [215,216,217]. One such HCV VLP vaccine, which contained HCV structural protein core, E1, and E2, induced strong and broad humoral and cellular immune responses in mice, baboons, and chimpanzees [215]. These triple-antigen VLP vaccines resembled the putative HCV virions and provided protection against HCV infection. To increase the multivalence of the HCV vaccine, one group designed a mammalian liver cell-derived quadrivalent HCV VLP vaccine, which contained core, E1, and E2 structural proteins of genotypes 1a, 1b, 2a, and 3a [218]. The vaccine induced strong antibodies, nAbs, and memory B and T cell responses in vaccinated mice [216]. Neutralizing human monoclonal antibodies (HuMAbs), which targeted conserved antigenic domain B and D epitopes of the E2 protein, was strongly bound to the quadrivalent HCV VLP vaccine. The induced broad humoral and cellular immune responses might be from the critical epitopes display ability of the VLP vaccine. Apart from extracellular vesicle-based VLPs, retrovirus-derived VLPs also have been applied in HCV broad-spectrum vaccines. Pierre Garrone et al. designed an HCV E1/E2-pseudotyped VLP vaccine, which co-expressed Gag proteins of the Moloney murine leukemia virus (MLV) and the E1/E2 proteins of HCV [217]. The double-antigen VLP vaccine induced robust levels of anti-E1/E2 antibodies and nAbs in both mice and macaques. More importantly, the HCV 1a-derived double-antigen vaccine also induced nAbs, which could cross-neutralize other genotypes, including 1b, 2a, 2b, 4, and 5, demonstrating broad-spectrum properties of retrovirus-based VLP vaccines. The high genetic variability of the HCV genome poses a major challenge for the development of a preventive vaccine. To address this issue, Annette von Delft et al. designed a rhesus adenovirus vector-based HCV vaccine targeting conserved regions of multiple HCV genomes, demonstrating high immunogenicity with high-titer, broad-spectrum, and cross-reactive T cell responses in pre-clinical models [219]. Additionally, protein polymer-based self-assembling nanoparticles also have been used for HCV vaccine development [220]. Kwinten Sliepen et al. designed a bicomponent recombinant HCV glycoprotein nanoparticle vaccine by presenting permuted E2 and E1 (E2E1) immunogens on I53-50 protein nanoparticles, resulting in enhanced nAb responses [221]. They also generated a mosaic nanoparticle vaccine by presenting six different E2E1 immunogens on the same I53-50 nanoparticle. The resulting mosaic E2E1 nanoparticle vaccine-induced nAbs neutralized both vaccine-matched viruses and mismatched genotypes, suggesting superior cross-reactive immune responses against different HCV variants. The above nanoparticle vaccines achieved a broad spectrum by covalently conjugating multiple heterologous antigens, while another study rationally designed broad-spectrum vaccines by presenting homologous optimized E2 cores on self-assembling nanoparticles [222]. The authors reengineered the variable region 2 of E2 into a truncated form, which preserved the conserved neutralizing epitopes. These optimized E2 proteins were subsequently displayed onto either 24-mer ferritin or 60-mer E2p. These nanoparticles induced more effective nAbs, which neutralized both autologous and heterologous HCV genotypes.

Collectively, nanoparticle vaccines, including VLP and self-assembling protein polymers, have shown remarkable efficacy against both HBV and HCV, as well as their different mutants or genotypes. These nanoparticle vaccines merit being further evaluated for their safety and effectivity in clinical trials, thereby providing additional options for next-generation hepatitis virus vaccines.

**Table 4 vaccines-12-00030-t004:** The applications of nanoparticles as delivery systems or adjuvants in hepatitis vaccines.

Viruses	Vaccines	Nanoparticles	Size	Functions	Status	Ref.
Hepatitis viruses	HBsAg/HBcAg VLP	VLP	22 nm to 25 nm	Elicit both humoral and cellular immune responses	Pre-clinical	[205]
	S/preS1/preS2 VLP	VLP	22 nm	Achieve high rates of seroprotection by eliciting elevated nAb levels at low antigen doses	Approved (VBI Vaccines)	[208,209]
	preS1-ferritin	Ferritin	15 nm to 30 nm	Activate both T follicular helper cells and B cells and induce high levels of preventive and therapeutic nAbs	Pre-clinical	[210]
	HBsAg-MNP	PLGA	186.6 nm	Elicit persistent humoral immunity and enhance cellular immune response	Pre-clinical	[211]
	HBcAg-MPLA	PLGA	279 nm	Induce strong Th1-biased cellular immune responses with more IFN-γ production	Pre-clinical	[212]
	HCV core/E1/E2 VLP	VLP	40 nm to 60 nm	Induce robust and broad humoral and cellular immune responses in mice, baboons, and chimpanzees	Pre-clinical	[215]
	HCV core/E1/E2 VLP targeting genotypes 1a, 1b, 2a, and 3a	VLP	50 nm	Induce strong antibodies, nAbs, and memory B and T cell responses	Pre-clinical	[216,218]
	MLV-Gag-based E1/E2 HCV VLP	VLP	100 nm to 120 nm	Induce cross-neutralizing antibodies against wide ranges of genotypes	Pre-clinical	[217]
	Adenovirus-vectored HCV VLP	VLP	80 nm to 120 nm	Reinforce immunogenicity by eliciting high-titer, broad-spectrum, and cross-reactive T cell responses	Pre-clinical	[219]
	HCV E2E1-I53-50	I53-50	35 nm	Elicit significant nAb responses with cross-reactivity	Pre-clinical	[221]
	Optimized HCV E2 NP	Ferritin/E2p	24.5 nm/34.5 nm	Elicit more effective nAbs to potently neutralize both autologous and heterologous HCV genotypes	Pre-clinical	[222]

## 6. The Applications of Nanoparticles in Other Antiviral Vaccines

Apart from the above-mentioned anti-influenza, anti-coronavirus, anti-HIV, and anti-hepatitis virus nanoparticle vaccines, nanoparticles also have promoted the development of many other antiviral vaccines (Table 5). The Zika virus (ZIKV) is an enveloped single-stranded plus-stranded RNA virus with a spherical shape and a diameter of approximately 40–70 nm [223,224]. ZIKV belongs to the *Flavivirus* genus of the *Flaviviridae* family, which is mainly transmitted through mosquito bites, sexual contact, and blood transfusion. ZIKV infection poses different risks for different groups of people. People infected with ZIKV usually remain asymptomatic or experience common symptoms including rash, fever, muscle/joint pain, and headache. However, infection with the virus during pregnancy can lead to microcephaly and other congenital malformations in infants [225]. There is currently no clinical vaccine available for preventing ZIKV infections. Ferritin nanoparticles have been used to develop vaccines against ZIKV. Aryamav Pattnaik et al. constructed a candidate vaccine with ferritin nanoparticles exhibiting Domain III (DIII) of ZIKV E protein [226]. Compared with immunogen monomers, the ferritin-based nanoparticles induced higher nAb responses and were capable of eliciting cell-mediated immune responses to eliminate virus-infected cells. Importantly, nanoparticle vaccine-induced nAbs could protect mice from lethal ZIKV challenges and potentially neutralize other heterologous ZIKV lineages. Haibin Hao et al. encapsulated live ZIKV within a chitosan oligomer hydrogel with built-in calcium carbonate nanoparticles (nano-CaCO_3_) as stabilizers and sources of Ca^2+^ [227]. This virus-entrapped composite hydrogel was named Vax. Trapped viruses were directly converted into antigens. Additionally, the self-adjuvant properties of chitosan scaffolds and nano-CaCO_3_ directly activated innate immunity by the activation of pattern recognition receptors (PRRs). The hydrogel-generated local inflammatory niche promoted the recruitment of immune cells, including granulocytes, macrophages, DCs, T cells, B cells, and NK cells. The potent activation of innate immune responses further induced robust adaptive immunity, characterized by stronger specific serum IgG responses and cellular immune responses, providing protection against fatal ZIKV infection. In addition, the Vax vaccine also demonstrated long-term protective effects through durable immune memory.

The Dengue virus (DENV) is another enveloped single-stranded plus-stranded RNA virus. DENV is small in size, measuring approximately 40–60 nm in diameter. Similar to ZIKV, DENV also belongs to the *Flavivirus* genus of the *Flaviviridae* family. There are four major serotypes of DENVs, including DENV-1, DENV-2, DENV-3, and DENV-4. All of these serotypes can infect humans, among which the severity rate and fatality rate of DENV-2 are higher than those of other types [12]. Symptoms associated with DENV infection include fever, muscle pain, severe headache, orbital pain, anorexia, and nausea. Dengue fever is an acute insect-borne infectious disease caused by DENV that threatens nearly 3.9 billion people worldwide. Dengue fever is one of the leading causes of death among children in Southeast Asia [228]. The Dengue vaccine Dengvaxia, developed by Sanofi Pasteur, was approved for market in the United States in 2019. Dengvaxia is a live attenuated quadrivalent vaccine composed of DENV components and the yellow fever virus (YFV) 17D vaccine. However, its protective efficacy is highly controversial [229,230,231]. Quang Huy Quach et al. designed different-sized gold nanoparticles (AuNPs), which were coated by DENV-2 domain III of envelop proteins (EDIII) (Table 5) [232]. The resulting AuNP-E vaccine induced specific anti-EDIII antibodies in AuNP core size in concentration-dependent manners. Additionally, AuNP-E vaccines stimulated cellular immune responses by inducing the proliferation of IFN-γ and IL-4-producing T cells, leading to the generation of nAbs that bound to EDIII proteins and authentic viruses. Moreover, AuNP accumulated less in major organs and demonstrated minimal toxicological and side effects. Another study loaded UV-inactivated DENV-2 onto *N*, *N*, and *N*-trimethyl chitosan nanoparticles (TMC NPs) and developed a self-adjuvanted vaccine [233]. The vaccine enhanced the immunogenicity of inactivated DENV-2 and promoted the maturation of monocyte-derived dendritic cells (MoDCs). Both Th1 and Th2 types of immune responses were driven by DENV-2 TMC NP vaccines, resulting in the differentiation and activation of DENV-2-specific cytotoxic T cells, as well as the increase in potent nAbs. The nano-vaccine strategies may hold promise for developing new vaccines targeting all serotypes of DENV and ZIKV, for which preventive measures are currently lacking.

Respiratory syncytial virus (RSV) is not only a leading cause of hospitalization for lower respiratory tract infections in infants but also a major contributor to severe respiratory diseases in the elderly [234]. RSV-associated acute lower respiratory tract infections accounted for approximately 59,600 pediatric deaths in children under the age of 5 in 2015 [235]. In recent years, RSV vaccine research has rapidly expanded, encompassing particle-based vaccines, attenuated vaccines, subunit vaccines, and carrier-based vaccines [236]. Two RSV vaccines have been approved by the FDA for preventing lower respiratory tract diseases caused by RSV among individuals aged over 60 years. The first approved RSV vaccine is named Arexvy, which is an RSV prefusion F protein-based vaccine developed by GSK [237,238]. The other is named Abrysvo, developed by Pfizer, which is a bivalent RSV prefusion F vaccine targeting both RSV A and RSV B [239]. To increase the antigen density, Jessica Marcandalli et al. designed a self-assembling protein nanoparticle vaccine that exhibited repetitive arrays of the prefusion F glycoprotein trimer (DS-Cav1) (Table 5) [240]. Twenty copies of DS-Cav1 trimers were displayed on I53-50 nanoparticles. The resulting DS-Cav1-I53-50 nanoparticles elicited more potent nAb responses and cellular responses than trimeric DS-Cav1 in immunized mice and non-human primates. RSV VLP vaccines also show advantages in displaying multiple copies of antigens or antigenic epitopes. A double-blind, placebo-controlled, dose–escalation study evaluated the safety and immunogenicity of a specific site-directed chemically defined RSV VLP vaccine named V-306 [241]. V-306 displayed 60–90 copies of RSV F site II protein mimetics (FsIIm) as antigenic epitopes on the surface of VLP. The clinical study demonstrated that this VLP vaccine was safe and well-tolerated for women aged 18–45 years. The FsIIm-specific IgG titers significantly increased and lasted over 4 months, representing its advantage in providing long-term protection against RSV infection.

The hemorrhagic fever and multiple organ failure caused by the Ebola virus (EBOV) have resulted in a high mortality rate [242]. Currently, there is only one EBOV vaccine that has been approved by the FDA, which is named ERVEBO or rVSVΔG-ZEBOV-GP. It is a replication-competent and attenuated recombinant vesicular stomatitis virus (rVSV)-vectored vaccine, which displays EBOV envelope glycoproteins [243]. EBOV nanoparticle vaccines are also under development and evaluation. Kelly L. Warfield et al. designed VLP-based vaccines by co-expressing EBOV glycoprotein (GP), matrix protein (VP40), and/or nucleoprotein (NP) in mammalian cells (Table 5) [244,245]. These vaccines induced high titers of EBOV-specific nAbs, which protected mice and cynomolgus macaques against lethal EBOV challenge. The above VP40-containing VLPs resemble filamentous infectious virions and are heterogeneous in shape and size, which may hinder the production and purification of VLPs. Another study designed an HIV-Gag-incorporated VLP vaccine, which formed smaller spherical nanoparticles [246]. These spherical VLPs could be more efficiently targeted to lymph nodes and captured by APCs. The authors co-expressed HIV Gag proteins with both Zaire Ebola virus (EBOV) GPs and Sudan Ebola virus (SUDV) GPs to produce the bivalent EBOV/SUDV GP-Gag VLPs. Vaccination of rhesus macaques with this bivalent VLP vaccine induced high titers of broad-spectrum nAbs that neutralized all four pathogenic Ebola viruses, as well as strong cellular immune responses. Traditional VLPs produced by mammalian cells are prone to degradation upon delivery into the host. To address this issue, James J. Moon et al. designed interbilayer-crosslinked multilamellar vesicles (ICMVs) as synthetic vaccine carriers, which released entrapped antigens at a slower rate [247]. Upon incorporating ICMVs with recombinant EBOV glycoproteins (rGPs), epitopes and quaternary structures of rGPs were properly maintained [248]. These rGP-ICMV VLPs elicited robust nAbs responses in mice, represented by efficiently generating GC B cells and polyfunctional T cells. Native EBOV GP proteins exist as trimers that meditate cell entry and induce bNAbs. Linling He et al. rationally designed thermostable native-like GP trimers and conjugated them onto 24-mer ferritin, 60-mer E2p, or 60-mer I3-01 [249]. These GP trimer-displayed self-assembling nanoparticle vaccines elicited cross-ebolavirus nAbs, demonstrating potential advantages of self-assembling protein nanoparticle in the development of universal EBOV vaccines. Saponin-based Matrix-M, the adjuvant of the COVID-19 nanoparticle vaccine NVX-CoV2373, has been demonstrated to enhance the immunogenicity of SARS-CoV-2 S trimer vaccines [162]. Prior to the outbreak of COVID-19, this nano-adjuvant has already been successfully applied in EBOV vaccines. Matrix-M-formulated EBOV GP nanoparticle vaccines elicited robust and persistent B cell and T cell immune responses, which were evidenced by higher titers of nAbs and IFN-γ production [250,251]. These nanoparticle vaccines provided full protection against lethal EBOV challenge in mice and were well-tolerated in healthy adults based on the results of a Phase I clinical trial [250]. Collectively, all the results of these pre-clinical studies and clinical trials demonstrated that nanoparticle vaccines could be applied in human populations to prevent pathogenic viral infections.

Nanoparticle vaccines represent a cutting-edge protein vaccine technology that offers a synergistic enhancement of both safety and efficacy. They are expected to become a rising star in the vaccine industry. Nevertheless, the development of nanoparticle vaccines is currently in its early stages, necessitating more clinical data to substantiate their safety profile and inherent advantages. At the same time, additional explorations are warranted in antigen selection and optimization strategies, as well as refining protein production processes.

**Table 5 vaccines-12-00030-t005:** The applications of nanoparticles as delivery systems or adjuvants in vaccines of other pathogenic viruses.

Viruses	Vaccines	Nanoparticles	Size	Functions	Status	Ref.
Other viruses	ZIKV E-ferritin	Ferritin	14 nm to 15 nm	Induce high nAb responses and protect mice from lethal ZIKV challenges	Pre-clinical	[226]
	Live ZIKV-entrapped hydrogel (Vax)	Chitosan/CaCO_3_	200 nm	Directly activate innate immunity by the activation of pattern recognition receptors (PRRs) and generate local inflammatory niche	Pre-clinical	[227]
	DENV-2 E-AuNP	AuNP	24.8 nm to 83.0 nm	Induce the proliferation of IFN-γ and IL-4-producing T cells and lead to the generation of potent nAbs	Pre-clinical	[232]
	Inactivated DENV-2 TMC NP	*N*, *N*, and *N*-trimethyl chitosan (TMC)	483.5 nm	Induce both Th1 and Th2 types of immune responses and increase activated cytotoxic T cells and potent nAbs	Pre-clinical	[233]
	RSV DS-Cav1-I53-50	I53-50	44 nm	Induce more potent nAb responses and cellular responses than trimeric DS-Cav1	Phase I	[240]
	RSV FsIIm VLP (V-306)	VLP	25 nm to 30 nm	Induce well-tolerated FsIIm-specific antibodies, which persist over 4 months	Phase I	[241]
	EBOV GP/VP40/NP VLP	VLP	70 nm to 150 nm	Induce high titers of EBOV-specific nAbs, which protect mice and cynomolgus macaques against lethal EBOV challenges	Pre-clinical	[244,245]
	EBOV/SUDV GP-Gag VLP	VLP	110 nm to 130 nm	Induce high titers of broad-spectrum nAbs that neutralize all four Ebola viruses	Pre-clinical	[246]
	rGP-ICMV VLP	Interbilayer-crosslinked multilamellar vesicle (ICMV)	117 nm	Efficiently generate GC B cells and polyfunctional T cells and elicit robust nAbs responses	Pre-clinical	[248]
	EBOV GP-ferritin/E2p/I3-01	Ferritin/E2p/I3-01	34.6 nm/45.9 nm/49.2 nm	Preserve thermostable native-like GP trimers and elicit cross-ebolavirus nAbs	Pre-clinical	[249]
	EBOV GP vaccine with Matrix-M	Matrix-M	40 nm	Elicit robust and persistent B cell and T cell immune responses, characterized by higher titers of nAbs and IFN-γ production	Phase I	[250,251]

## 7. Conclusions

Although numerous vaccines have been developed to combat pathogenic viruses, most vaccines are still in the developmental stage. With the outbreak of COVID-19, many different kinds of vaccines show their advantages in preventing SARS-CoV-2 infection. However, achieving complete protection against viral infections, particularly caused by viral mutants, remains a big challenge. High titers, broad spectrums, long-term persistence, and protective immunity are the four major goals of antiviral vaccines (Figure 4). With the development of novel antigen presentation strategies, most vaccines can elicit sufficient nAb titers to neutralize invading viruses. However, as viruses can accumulate escape mutations, vaccines derived from original viruses are often unable to induce nAbs to bind or neutralize viral mutants. Thus, how to design an efficient vaccine that can confront early wild type viruses and later viral mutants represents a crucial issue. Additionally, it is worth noting that virus pandemics, especially respiratory virus pandemics, may seasonally reappear. Therefore, designing vaccines that are capable of providing long-term protection will be another important research direction. The toughest difficulty of developing vaccines lies in eliciting protective immunity. Based on vaccine effectiveness, protective immunity can be categorized into four progressive levels: preventing death, preventing severe disease, preventing mild disease, preventing transmission, and preventing infection. For confronting SARS-CoV-2, most vaccines can efficiently prevent death and severe diseases, while few vaccines can prevent mild diseases, transmissions, or even infections.

To solve the above issues, we need to fully elucidate the structural basis of antigens for eliciting bNAbs. Glycoproteins of many viruses exist as trimers on the surfaces of viruses. Maintaining the structure of native-like trimers upon designing immunogens is important for eliciting bNAbs, which target diverse viral variants. To generate nAbs that persist for longer times, we need to design vaccines or vaccination strategies that can induce higher titers of antibodies and more antigen-specific memory B cells (MBCs) and long-lived plasma cells (LLPCs). However, to design preventive vaccines that can provide protection against respiratory viruses, it is necessary to re-think the immunological underpinnings of innate immunity, adaptive immunity, and mucosal immunity. Regardless of vaccine type, the produced immunogens are recognized and internalized by antigen-presenting cells (APCs), including macrophages, dendritic cells (DCs), and B cells, followed by digesting and presenting to T cells. This process initiates and activates adaptive immune responses by promoting the activation, proliferation, and differentiation of both antigen-specific T cells and B cells. As we have mentioned above, nano-adjuvants, including gold nanoparticles (AuNPs), gold nanorods (AuNRs), and Saponin-based Matrix-M, can facilitate the uptake of antigens by APCs, resulting in enhanced antigen presentation [98,99,162,197,232,250,251]. Future investigations should consider employing more effective adjuvants, such as nano-adjuvants, to strengthen the antigen presentation function of APCs. Additionally nano-adjuvants, nano-carriers capable of tightly trapping and slowly releasing antigens, make APCs continuously receive the stimulation of immunogens. This controlled-release procedure is crucial for the maturation of antibody-secreting plasma cells and the production of corresponding high-affinity nAbs. Another limiting factor that impairs the cross-talk of innate immunity and adaptive immunity is the lymph node targeting of antigens. Both antigen-displayed virus-like particles (VLPs) and self-assembling protein nanoparticles possess similar shape and size characteristics as viruses. Upon being administrated into the host, these nanoparticles can quickly drain into lymph nodes and be captured by APCs. Within the germinal centers (GCs) of lymph nodes, nanoparticle vaccines can also be captured by follicular dendritic cells (FDCs). FDCs directly present nanoparticles to B cells without further digestion, thereby preserving the spatial conformation of antigens and inducing bNAbs targeting highly conserved epitopes.

The biggest challenge in developing preventive respiratory virus vaccines lies in comprehensively understanding mucosal immunity and designing mucosal vaccines. The mucosal immune system is the first line of defense against respiratory viruses, such as influenza viruses and coronaviruses. Antigens are captured by epithelial cells or microfold cells (M cells). These cells transport and deliver antigens to APCs for further antigen presentation. Activated APCs then present antigens to T cells, priming and activating B cell immune responses. The major adaptive humoral immune response within mucosal surfaces is mediated by secretory IgA (sIgA) nAbs, while locally produced antigen-specific IgM and IgG also contribute significantly to the mucosal defense against corresponding viruses [252]. The local immunity within the upper respiratory tract (URT) and the lower respiratory tract (LRT) of mammalians occurs in various mucosa-associated lymphoid tissues (MALTs), including Waldeyer’s ring, nasal-associated lymphoid tissues (NALTs), bronchus-associated lymphoid tissues (BALTs), and inducible bronchus-associated lymphoid tissues (iBALTs) [253]. However, intramuscularly administrated vaccines often induce strong serum-derived IgG-committed immune responses instead of sIgA humoral defense, resulting in delayed virus neutralization in respiratory tracts. Thus, designing mucosal vaccines that are administrated intranasally can potentially elicit more MALT-derived sIgA and IgG nAbs. In addition to optimizing vaccination routes, mucosa-targeting adjuvants and carriers play crucial roles in facilitating the transport and presentation of antigens. The adenosine diphosphate (ADP)-ribosylating bacterial enterotoxins cholera toxin (CT) and *E. coli.* heat-labile toxins (LTs) are the most potent mucosal adjuvants to date [254]. They directly activate B cells without the assistance of CD4^+^ T cells. However, these enterotoxins are also highly toxic. Future work needs to develop their derivatives to reduce their toxicity while maintaining their effectiveness. Nanoparticle-based adjuvants and carriers have shown superior advantages in nasal vaccination [255]. Chitosan is a positively charged polymer nanoparticle that can easily bind to negatively charged mucosa and epithelial cells and has been widely used to develop mucosal vaccines. Inactivated vaccines, DNA vaccines, and protein subunit vaccines, which are adjuvanted with chitosan, have been found to induce strong sIgA-committed mucosal immunity upon being administrated intranasally [85,172,256]. Similarly, PLGA polymer nanoparticles are also promising mucosal vaccine carriers that facilitate the induction of mucosa-resident nAbs, as well as protective CD8^+^ memory T cells [86,257]. Recently, mucosa-targeting self-assembling protein nanoparticle vaccines without any additional adjuvants have been applied in influenza vaccines. Ferritin-based influenza nanoparticle vaccines induced cross-protective sIgA nAb responses and T cell immune responses, demonstrating their potential in developing universal influenza vaccines [92]. More nanoparticles merit being further evaluated in the induction of mucosal immunity and clinical applications of mucosal vaccines.

To design mucosal vaccines, particularly nanoparticle-based vaccines, several challenges should be carefully resolved. Upon depositing on the mucosal surface, vaccines first need to cross the physical layer of airway mucus, which is mainly produced by secretory cells. Antimicrobial molecules, immunomodulatory molecules, and protective molecules within mucus gel layers can physically or enzymatically eliminate adherent pathogens or vaccines [258]. Additionally, apical cilia on ciliated cells can rhythmically beat to promote antigen-deposited mucus motility, the process of which is called mucociliary clearance (MCC) [259]. After crossing mucus, antigens need to be recognized and transported by M cells, which efficiently capture particles and macromolecules [260]. Consequently, nanoparticle vaccines can be more easily captured and transported by M cells than monomer vaccines. Mucosa-targeting adjuvants can boost mucosal immune responses. However, some adjuvants can be transported from the olfactory tissues into the central nervous system (CNS) [261]. Future endeavors need to focus on developing safer delivery systems and adjuvants that can be intranasally administrated and targeted to MALTs without brain deposition. Another potential disadvantage of protein-based vaccine platforms lies in the pre-existing immunity to these self-assembling protein nanoparticles themselves. Several reports have shown that protein nanoparticles or VLPs were immunogenic and antigenic, accompanied by both T cell and B cell immune responses against these nanoparticles [88,95,147]. However, these reports also showed that the induction of nAbs against conjugated antigens or later immunized heterologous antigens was not influenced or diminished. Further investigation is still required to determine whether frequent immunization with immunogenic nanoparticles could impede the induction of antigen-specific antibodies. As nanoparticle vaccines are just beginning to emerge, their safety and effectiveness need to be carefully evaluated in the next few decades.

## Figures and Tables

**Figure 1 vaccines-12-00030-f001:**
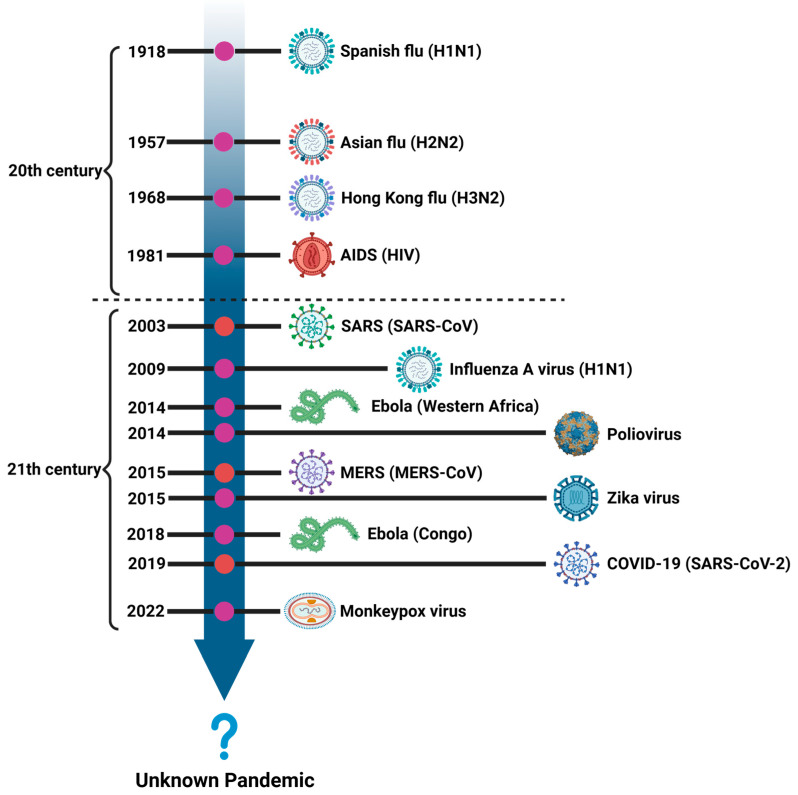
Major pathogenic viruses and corresponding pandemics. In 1918, the Spanish flu was caused by H1N1. In 1957, the Asian flu was caused by H2N2. In 1968, the Hong Kong flu was caused by H3N2. In 1981, AIDS was caused by HIV. In 2003, SARS-CoV caused SARS. In 2009, H1N1 caused another influenza A pandemic. In 2014, Western Africa experienced Ebola hemorrhagic fever. The poliovirus pandemic also took place in 2014. In 2015, MERS-CoV caused MERS, and a Zika virus pandemic also occurred. In 2018, another Ebola pandemic happened in Congo. In 2019, SARS-CoV-2 caused COVID-19. In 2022, the monkeypox viruses started to spread among human society. In the future, more unknown viral pandemics may occur.

**Figure 2 vaccines-12-00030-f002:**
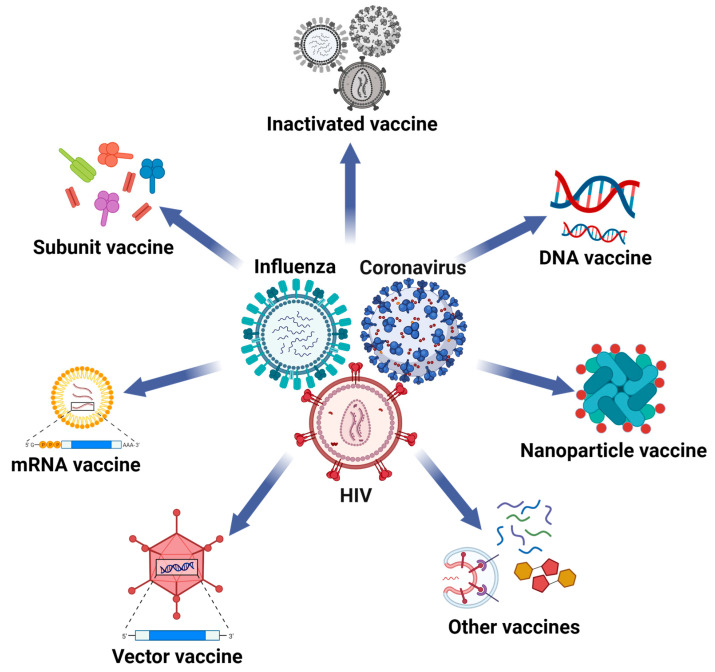
Different types of antiviral vaccines. Many different kinds of vaccines have been developed to combat influenza viruses, coronaviruses, and HIV, which include inactivated vaccines, DNA vaccines, mRNA vaccines, protein subunit vaccines, viral vector-based vaccines, nanoparticle vaccines, and many other types.

**Figure 3 vaccines-12-00030-f003:**
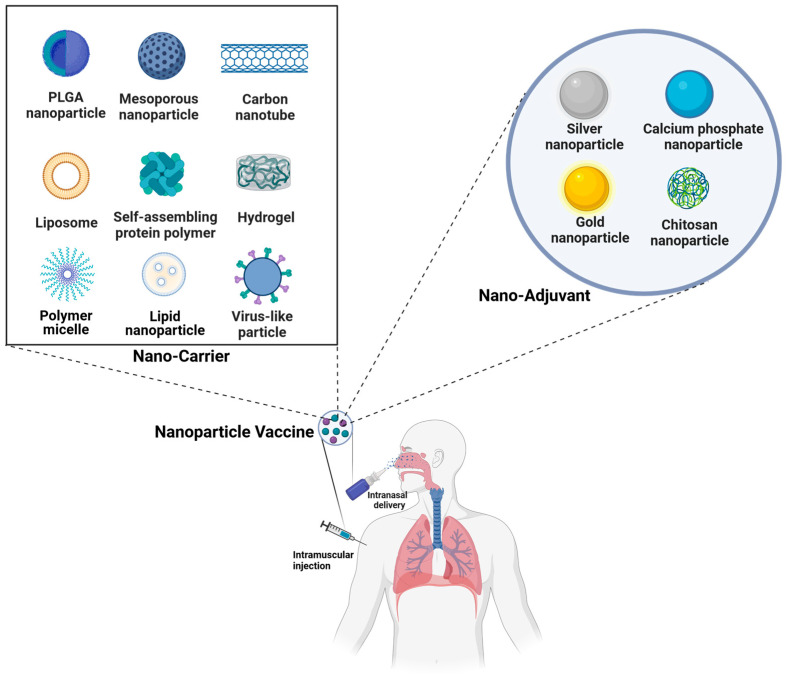
Nanoparticles as vaccine carriers and adjuvants. Nanoparticles can be utilized as both immunogen carriers and vaccine adjuvants. Nano-carriers include PLGA nanoparticles, mesoporous nanoparticles, carbon nanoparticles, liposome-based nanoparticles, self-assembling protein polymer-based nanoparticles, lipid-based nanoparticles, virus-like particles, hydrogels, and polymer micelles. Both organic and inorganic nanoparticles can be used as nano-adjuvants, which include silver nanoparticles, gold nanoparticles, calcium phosphate nanoparticles, and chitosan nanoparticles.

**Figure 4 vaccines-12-00030-f004:**
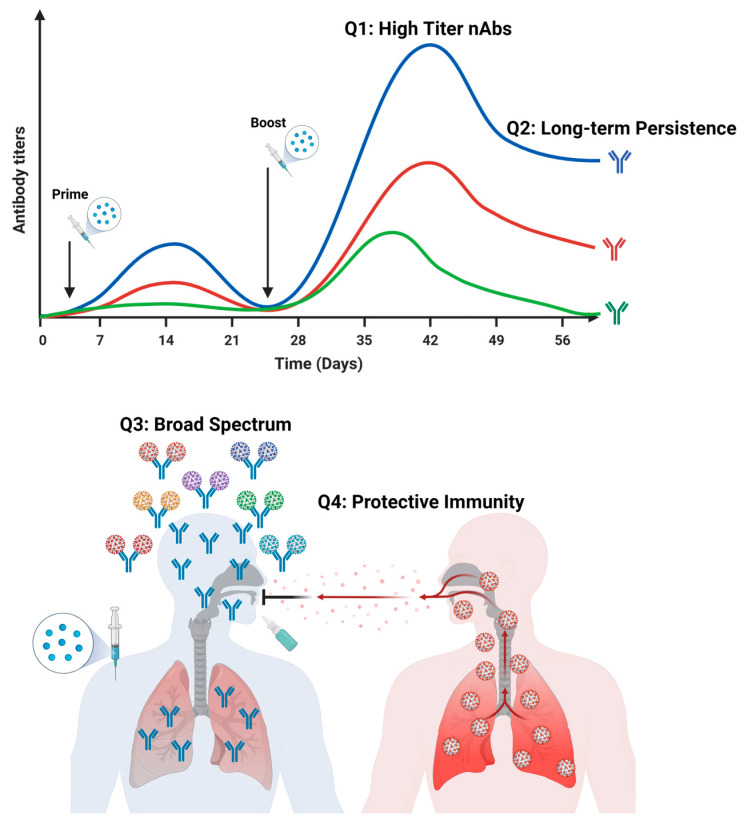
Four major questions or challenges of antiviral vaccines. The first challenge of antiviral vaccines is to induce high titers or at least sufficient titers of nAbs capable of quickly neutralizing invaded viruses. The second challenge lies in the long-term persistence of induced antibodies. Most vaccines fail to provide long-term protection, as the induced nAbs decay quickly within several months. The third question revolves around designing a universal vaccine that induces broad-spectrum nAbs. Most types of viruses acquire escape mutations under the selective pressure of immune responses, which compromises the effectiveness of original virus-derived vaccines. Thus, it is essential to develop universal vaccines that can provide protection against both autologous viruses and corresponding mutants. Lastly, the fourth challenge is to provide efficient protective immunity, particularly in terms of preventing transmission and preventing infection.

## Data Availability

Data are contained within the article.

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
