# Peer review of "Nanoparticles and Antiviral Vaccines"

_vaccines, 2023, doi:10.3390/vaccines12010030_

Round 1

Reviewer 1 Report

Comments and Suggestions for Authors

The paper by Sen Liu et. al. entitled “Nanoparticles and Antiviral Vaccines” is a very well written and interesting review. Authors described different kind of vaccines against the most dominant and dangerous viruses causing pandemics and epidemics. They summarized different antiviral vaccines and adjuvants. The most valuable part of manuscript concerns recent advances of next-generation vaccines in terms of application of nanoparticles. It contains many detailed and valuable and up to date information. My recommendation is to accept manuscript after minor revision. In details my suggestions are below:

1/ In manuscript it is not always clearly presented which vaccine is approved for use or on which step of clinical trial they are. Such information should be added in tables with lists of vaccines (as additional column). The name of company for approved vaccines would be also very informative.

2/ Line 175-176. Word “binding” in this sentence is not adequate to information authors would like to provide. Please, write this sentence again.

3/ line 253-254. There are more subtypes of influenza A virus. Maybe authors wanted to convey different information? Please correct this sentence to be accurate.

4/ Authors contribution statement lines. “Validation” and “formal analysis” terms belong to experimental papers and are not adequate to review paper. I recommend to delete these and leave only necessary descriptions of authors’ contributions.

Author Response

Reviewer #1:

The paper by Sen Liu et. al. entitled “Nanoparticles and Antiviral Vaccines” is a very well written and interesting review. Authors described different kind of vaccines against the most dominant and dangerous viruses causing pandemics and epidemics. They summarized different antiviral vaccines and adjuvants. The most valuable part of manuscript concerns recent advances of next-generation vaccines in terms of application of nanoparticles. It contains many detailed and valuable and up to date information. My recommendation is to accept manuscript after minor revision. In details my suggestions are below:

Reply: We appreciate for reviewer’s comprehensive summaries and giving us the following insightful suggestions. We have carefully reviewed every comment below and accordingly revised the manuscript.

  1. In manuscript it is not always clearly presented which vaccine is approved for use or on which step of clinical trial they are. Such information should be added in tables with lists of vaccines (as additional column). The name of company for approved vaccines would be also very informative.

Reply: We sincerely apologize for omitting this important information in our original manuscript. We have added the clinical status of each vaccine, and the company name of the approved vaccine to all tables. This information has been shown in additional column as the reviewer kindly suggested.

  1. Line 175-176. Word “binding” in this sentence is not adequate to information authors would like to provide. Please, write this sentence again.

Reply: We thank the reviewer for pointing out this omission and we have modified the sentence. As another reviewer has kindly reminded, the likelihood of adenovirus and vesicular stomatitis virus vectors integrating into the host genome is exceedingly low. To avoid overstatement and inaccuracy, we have deleted this controversial sentence.

  1. line 253-254. There are more subtypes of influenza A virus. Maybe authors wanted to convey different information? Please correct this sentence to be accurate.

Reply: We thank the reviewer for the constructive suggestion. We have corrected the sentence in lines 253-254. The revised sentence is shown as below: “Based on combinations of viral surface proteins hemagglutinin (HA) and neuraminidase (NA), influenza A viruses contain H1N1, H3N2, H5N1, H7N9 and many other subtypes.”

  1. Authors contribution statement lines. “Validation” and “formal analysis” terms belong to experimental papers and are not adequate to review paper. I recommend to delete these and leave only necessary descriptions of authors’ contributions.

Reply: We thank the reviewer for the kind suggestions. We have deleted “Validation” and “formal analysis” terms and only kept the necessary description in the Author Contributions section.

Reviewer 2 Report

Comments and Suggestions for Authors

Despite the fact that the review is generally well-structured and will be interesting to readers, I have several questions and comments:

General comments:

-The term “viral-like particles” should be replaced with more common term “virus-like particles”.

-It would be interesting for readers if information on the level of immune response to the protein-based platforms (ferritin, PapMV VLPs, etc.) were provided in places where these platforms/adjuvants are discussed.

-Due to the fact that the review is devoted to the topic of nanoparticles, it would be appropriate to add a column with information about size to all tables.

Point-by-point comments:

-Lines 114-115. “However, specific available drugs against viruses are still lacking, except for HCV”. And how about drugs against herpesviruses and retroviruses? Probably, the phrase should be rephrased and emphasized in it that drugs against HCV allow the organism to get rid of the virus.

-Line 120. It is better to replace the word “sterilized” with “treated”.

-Lines 122-125 and Lines 888-890. Despite the fact that the process of preparing plasma-derived hepatitis B vaccine includes the stage of inactivation, I suppose it is not correct to classify this vaccine as "inactivated". The main component of this vaccine is exactly HBsAg, not the whole virion. In this case, inactivation was required because the source of HBsAg was human blood, which could additionally contain infection virions of HBV (not obligatory) and other known and unknown pathogens. The major vaccine component (HBsAg) itself does not require inactivation. So, classification of plasma-derived vaccine against HBV as "inactivated" confuses the reader. Moreover, within the Introduction (line 125) it is difficult to figure out that the authors implied exactly the plasma-derived vaccine since the yeast-produced HBsAg-based HBV vaccine is very well-known as the first recombinant vaccine. I recommend avoiding the classifying of any of this HBV vaccine as "inactivated".

-Line 176. “…the risk of binding to the human host genome…” This is a rather brave thesis that requires citing not other reviews, but original articles. Furthermore, the review by Bulcha et al., 2021 (reference 49) gives rather the opposite thesis. Here are the citations from the review by Bulcha et al., 2021:

In the nucleus, the viral DNA predominantly remains epichromosomal and is not incorporated into the host cell genome”;

“Ad vectors have seen a rebirth in human gene therapy research. They maintain many practical advantages, including their broad tropism profiles, lack of host genome integration, and large packaging capacities (~36 kb)”.

-Lines 184-185. The part of the phrase “demonstrating extraordinary effectiveness in virus prevention” should be replaced with “demonstrating virus prevention effect”. “Extraordinary effectiveness” is a controversial statement for recombinant vaccines.

-Line 310. “TLR7/8a-conjugated” should be replaced with “TLR7/8 agonist-conjugated”.

-Line 329. “Helicobacter pylori” should be written in italics.

-Information in the Introduction and in Lines 478-486 intersects quite strongly. With a considerable volume of the MS text, authors should look at how appropriate it is to duplicate information.

-Line 990. It seems that it would be better to use "many other antiviral vaccines" or "other antiviral vaccines" than “more other antiviral vaccines”.

-Line 1018. The size of DENV virions can’t be 17-25 nm. The authors should check the information on the size of the virion of this virus.

-Line 1040, Table 5. “both Th1-biased and Th2-biased cellular responses”

The immune response can be either Th1-biased or Th2-biased. It is better to write “both Th-1 and Th-2 types of immune response”. 

Author Response

Reviewer #2:

Despite the fact that the review is generally well-structured and will be interesting to readers, I have several questions and comments:

Reply: We appreciate the positive feedback and constructive comments of the reviewer, as well as the effort and time required to review our manuscript. We have addressed each comment according to the reviewer’s suggestion.

General comments:

-The term “viral-like particles” should be replaced with more common term “virus-like particles”.

Reply: We thank the reviewer for pointing out this omission. We have replaced the word “viral-like particles” with “virus-like particles” within the whole manuscript as well as figures.

It would be interesting for readers if information on the level of immune response to the protein-based platforms (ferritin, PapMV VLPs, etc.) were provided in places where these platforms/adjuvants are discussed.

Reply: We thank the reviewer for the suggestion. Protein-based platforms (ferritin, PapMV VLPs) may contain potential immunogenicity of their own, thus influencing the effectiveness of the vaccine. We have read all the related papers and discussed this potential disadvantage within the Discussion section as below: “Another potential disadvantage of protein-based vaccine platforms lies in the pre-existing immunity to these self-assembling protein nanoparticles themselves. Several reports have shown that protein nanoparticles or VLPs were immunogenic and antigenic, accompanied by both T-cell and B-cell immune responses against these nanoparticles [88, 95 and 147]. However, these reports also showed that the induction of nAbs against conjugated anti-gens or later immunized heterologous antigens was not influenced or diminished. Further investigation is still required to determine whether frequent immunization with immunogenic nanoparticles could impede the induction of antigen-specific antibodies.”

Due to the fact that the review is devoted to the topic of nanoparticles, it would be appropriate to add a column with information about size to all tables.

Reply: We thank the reviewer for the insightful comment and providing sound advice. We have added information about nanoparticle sizes within all tables, which were presented within additional columns.

Point-by-point comments:

Lines 114-115. “However, specific available drugs against viruses are still lacking, except for HCV”. And how about drugs against herpesviruses and retroviruses? Probably, the phrase should be rephrased and emphasized in it that drugs against HCV allow the organism to get rid of the virus.

Reply: We are sorry for this confusing sentence. We have re-written this sentence as below: “However, most drugs fail to eradicate viruses or provide long-term protection. While drugs against HCV have been able to allow the organism to get rid of the virus.”

Line 120. It is better to replace the word “sterilized” with “treated”.

Reply: Thank you for your kind remind. We have replaced “sterilized” with “treated” in our revised manuscript.

-Lines 122-125 and Lines 888-890. Despite the fact that the process of preparing plasma-derived hepatitis B vaccine includes the stage of inactivation, I suppose it is not correct to classify this vaccine as "inactivated". The main component of this vaccine is exactly HBsAg, not the whole virion. In this case, inactivation was required because the source of HBsAg was human blood, which could additionally contain infection virions of HBV (not obligatory) and other known and unknown pathogens. The major vaccine component (HBsAg) itself does not require inactivation. So, classification of plasma-derived vaccine against HBV as "inactivated" confuses the reader. Moreover, within the Introduction (line 125) it is difficult to figure out that the authors implied exactly the plasma-derived vaccine since the yeast-produced HBsAg-based HBV vaccine is very well-known as the first recombinant vaccine. I recommend avoiding the classifying of any of this HBV vaccine as "inactivated".

Reply: We apologize for this confusion. These HBV vaccines indeed cannot be classified as inactivated vaccines and we have deleted all of these wrong sentences.

Line 176. “…the risk of binding to the human host genome…” This is a rather brave thesis that requires citing not other reviews, but original articles. Furthermore, the review by Bulcha et al., 2021 (reference 49) gives rather the opposite thesis. Here are the citations from the review by Bulcha et al., 2021:

“In the nucleus, the viral DNA predominantly remains epichromosomal and is not incorporated into the host cell genome”;

“Ad vectors have seen a rebirth in human gene therapy research. They maintain many practical advantages, including their broad tropism profiles, lack of host genome integration, and large packaging capacities (~36 kb)”.

Reply: We thank the reviewer for pointing out this improper description. Adenoviruses and vesicular stomatitis viruses’ vectors indeed unlikely have the risk of integrating within the host genome, which are potentially safe platform for vaccine delivery. To avoid overstatement and inaccuracy, we have deleted this controversial sentence in the revised manuscript.

Lines 184-185. The part of the phrase “demonstrating extraordinary effectiveness in virus prevention” should be replaced with “demonstrating virus prevention effect”. “Extraordinary effectiveness” is a controversial statement for recombinant vaccines.

Reply: We apologize for this overstatement. We have modified the sentence as follows: “Moreover, these vaccines are readily recognized by immune cells and effectively activate host immune responses, demonstrating virus prevention effect”.

Line 310. “TLR7/8a-conjugated” should be replaced with “TLR7/8 agonist-conjugated”.

Reply: Thank you for your kind remind. We have replaced the word “TLR7/8a-conjugated” with “TLR7/8 agonist-conjugated”.

Line 329. “Helicobacter pylori” should be written in italics.

Reply: Thanks for pointing out this mistake and we have corrected it.

Information in the Introduction and in Lines 478-486 intersects quite strongly. With a considerable volume of the MS text, authors should look at how appropriate it is to duplicate information.

Reply: We thank the reviewer for pointing out this omission. In our revised manuscript, we have simplified and re-written these sentences in Lines 478-486 as follows: “As we have mentioned previously, three pathogenic human coronaviruses can cause severe acute lung injury (ALI) or acute respiratory distress syndrome (ARDS), which include severe acute respiratory syndrome coronavirus (SARS-CoV), Middle East respiratory syndrome coronavirus (MERS-CoV) and SARS-CoV-2. Another four human coronaviruses including HCoV-OC43, HCoV-NL63, HCoV-229E and HCoV-HKU1 only lead to mild respiratory symptoms, while still can cause severe respiratory illness in the elderly and children”.

Line 990. It seems that it would be better to use "many other antiviral vaccines" or "other antiviral vaccines" than “more other antiviral vaccines”.

Reply: We thank the reviewer for this suggestion. We have replaced “more other antiviral vaccines” with “many other antiviral vaccines” in our revised manuscript.

Line 1018. The size of DENV virions can’t be 17-25 nm. The authors should check the information on the size of the virion of this virus.

Reply: We sincerely apologize for making this mistake. The particle diameter of DENV virions is about 40-60 nm. (PMID: 21219187; PMID: 24155405; PMID: 11893341).

Line 1040, Table 5. “both Th1-biased and Th2-biased cellular responses”

The immune response can be either Th1-biased or Th2-biased. It is better to write “both Th-1 and Th-2 types of immune response”.

Reply: We sincerely thank the reviewer for the kind suggestion. We have changed the original sentence as “Both Th1 and Th2 types of immune responses”. Similarly, we also have corrected this description within Table 5.

Reviewer 3 Report

Comments and Suggestions for Authors

The manuscript by Liu et al. delivers a timely review on the topics of nanoparticles and antiviral vaccines. This review systematically covers the history of major pathogenic viruses and pandemics, the development of various types of antiviral vaccines, and the role of nanoparticles in enhancing vaccine efficacy. The focus of the review is on the application of nanoparticles in the development of next-generation vaccines against a range of viruses, including influenza, coronaviruses, HIV, and hepatitis viruses. I will summarize my thoughts in its scientific aspects, relevance, contribution to the field, and English.  

  1. Depth of Content. The manuscript provides an extensive and detailed overview of the history of pathogenic viruses and the evolution of vaccine technologies. The focus on nanoparticles as a novel approach to vaccine development is timely and relevant.
  2. Scientific Accuracy and Relevance. The review appears to be scientifically accurate, citing numerous studies and clinical trials. The discussion on the potential of nanoparticles in vaccine development is particularly insightful, reflecting recent advances in the field.
  3. Innovation and Contribution to the Field. The manuscript contributes to the field by synthesizing a large body of research on nanoparticles in vaccine development. It highlights the potential of this technology to address current limitations in vaccine efficacy and immune response.
  4. Comments on English Language. The review is well-structured, with each section logically flowing into the next. The English grammar and syntax appear to be of reasonable standard, with no significant issues.

Overall, the manuscript is a comprehensive, well-written, and scientifically robust review of nanoparticles in antiviral vaccine development. It is an excellent resource for researchers and professionals in the field, providing a thorough overview of past and current developments, as well as future prospects in vaccine technology using nanoparticles. 

Author Response

Reviewer #3:

The manuscript by Liu et al. delivers a timely review on the topics of nanoparticles and antiviral vaccines. This review systematically covers the history of major pathogenic viruses and pandemics, the development of various types of antiviral vaccines, and the role of nanoparticles in enhancing vaccine efficacy. The focus of the review is on the application of nanoparticles in the development of next-generation vaccines against a range of viruses, including influenza, coronaviruses, HIV, and hepatitis viruses. I will summarize my thoughts in its scientific aspects, relevance, contribution to the field, and English.

  1. Depth of Content. The manuscript provides an extensive and detailed overview of the history of pathogenic viruses and the evolution of vaccine technologies. The focus on nanoparticles as a novel approach to vaccine development is timely and relevant.
  2. Scientific Accuracy and Relevance. The review appears to be scientifically accurate, citing numerous studies and clinical trials. The discussion on the potential of nanoparticles in vaccine development is particularly insightful, reflecting recent advances in the field.
  3. Innovation and Contribution to the Field. The manuscript contributes to the field by synthesizing a large body of research on nanoparticles in vaccine development. It highlights the potential of this technology to address current limitations in vaccine efficacy and immune response.
  4. Comments on English Language. The review is well-structured, with each section logically flowing into the next. The English grammar and syntax appear to be of reasonable standard, with no significant issues.

Overall, the manuscript is a comprehensive, well-written, and scientifically robust review of nanoparticles in antiviral vaccine development. It is an excellent resource for researchers and professionals in the field, providing a thorough overview of past and current developments, as well as future prospects in vaccine technology using nanoparticles.

Reply: We appreciate for reviewer’s positive feedback and comprehensive summaries, as well as the effort and time required to review our manuscript.

Again, we sincerely thank all the editors and reviewers for pointing out these important details. We have corrected the mistakes which the reviewers have reminded. We also have carefully read our revised manuscript many times to correct any typos and language issues. We hope that these changes are satisfactory.

Round 2

Reviewer 2 Report

Comments and Suggestions for Authors

The authors have addressed almost all points, and I recommend publication of the revised manuscript.